# 4-Aminoquinolines block heme iron reactivity and interfere with artemisinin action

**Melissa Rosenthal[1], Daniel E Goldberg[1,2]***

[1]Division of Infectious Diseases, Washington University School of Medicine, Saint Louis, United States; [2]Department of Molecular Microbiology, Washington University School of Medicine, Saint Louis, United States

## eLife Assessment

This is an **important** study with direct implications for the rational selection of antimalarial drug combinations. The authors present data demonstrating antagonism between 4-aminoquinoline anti-malarials and peroxide drugs under physiologically relevant conditions, including robust effects at the trophozoite stage and for chloroquine at the ring stage. While the conclusions are based on in vitro assays and further work will be needed to fully resolve the underlying mechanism, the findings are **convincing** and provide a strong rationale for evaluating drug combinations in relevant preclinical models prior to clinical testing.

***For correspondence:**
dgoldberg@wustl.edu

**Competing interest:** The authors declare that no competing interests exist.

## Abstract

Artemisinin-based combination therapies (ACTs) remain the mainstay of treatment for *Plasmodium falciparum* malaria, despite reports of ACT treatment failure. ACTs consist of an artemisinin and a longer-lived partner drug, which is often a quinoline. Given that heme is central to the mechanism of action of artemisinins and some quinolines, we hypothesized that these antimalarials would exhibit strong drug-drug interactions. Previous studies using standard 48 hr or 72 hr assays identified additive to mildly antagonistic interactions between artemisinins and quinolines. Here, we sought to re-evaluate these interactions using a pulsing assay that better mimics the short in vivo half-life of artemisinins. We found that chloroquine (CQ), piperaquine (PPQ), and amodiaquine substantially antagonize dihydroartemisinin (DHA), the active metabolite of artemisinins. CQ-DHA antagonism was notably exacerbated in CQ-resistant parasites, resulting in a superantagonistic phenotype in isobolograms. Further, we found that CQ co-treatment conferred artemisinin resistance to Kelch 13 wild-type parasites in the ring-stage survival assay. Using a small molecule probe (Ac-H-FluNox) to measure chemically reactive heme in live parasites, we determined that quinolines block artemisinin activation by rendering cytosolic heme inert. Finally, we probed beyond traditional ACTs, evaluating interactions of the proposed triple ACT, DHA-PPQ-mefloquine, as well as OZ439-quinoline combinations, which were all found to be antagonistic. Collectively, these in vitro data suggest that peroxide-quinolines may have liabilities as combination therapies.

## Introduction

Malaria was responsible for an estimated 263 million cases and 597,000 deaths in 2023, the majority of which were caused by *Plasmodium falciparum* (*World Health Organization, 2024*). The World Health Organization recommends artemisinin-based combination therapies (ACTs) for the treatment of *P. falciparum* malaria. ACTs are composed of a short-lived artemisinin derivative paired with a longer-lived partner drug. Depending on geographical resistance profiles, one of six combinations

is recommended: dihydroartemisinin (DHA)+piperaquine (PPQ), artesunate (AS)+mefloquine (MFQ), AS+amodiaquine (ADQ), AS+pyronaridine (PYR), artemether (AM)+lumefantrine (LM), or AS+sulfadoxine-pyrimethamine (SP). Three of these six partner drugs, PPQ, MFQ, and ADQ, possess quinoline groups. Importantly, heme is central to the mechanism of action of both artemisinins and quinolines (*Wicht et al., 2020*; *Rosenthal and Ng, 2020*).

Asexual blood-stage parasites are adept at thriving in an incredibly heme-rich environment. Parasites import up to 80% of host cell hemoglobin (*Sigala et al., 2015*), which is transported to a specialized lysosome-like organelle, termed the digestive vacuole. Here, hemoglobin is proteolytically processed into globin peptides and redox active heme ($Fe^{2+}$-ferroprotoporphyrin IX), which is oxidized to hemin ($Fe^{3+}$-ferroprotoporphyrin IX) (*Matz, 2022*). The majority of hemin is retained within the digestive vacuole where it is detoxified via sequestration into inert hemozoin crystals (*Egan et al., 2002*; *Combrinck et al., 2013*; *Zhang et al., 1999*). Hemin concentrations approach 0.5 M within the digestive vacuole (*Francis et al., 1997*). Blood-stage parasites also maintain a relatively high cytosolic heme concentration of approximately 1.6 µM (*Abshire et al., 2017*). While the source of this labile heme pool has not been experimentally determined, it is thought to originate from hemoglobin digestion.

The heme-rich environment of asexual blood stages has long been exploited for antimalarial intervention. The 4-aminoquinolines CQ and PPQ are thought to act primarily in the parasite digestive vacuole, where they inhibit hemozoin formation by binding heme/hemin and the growing face of hemozoin (*Sullivan et al., 1996*; *Sullivan et al., 1998*; *Olafson et al., 2017*; *Olafson et al., 2015*; *Adams et al., 1996*; *Egan et al., 1997*). It has been proposed that disruption of heme homeostasis and/or accumulation of toxic heme-quinoline complexes may contribute to parasite killing (*de Villiers and Egan, 2021*; *Fitch, 2004*; *Kapishnikov et al., 2019*). Resistance to 4-aminoquinolines is mediated by a collection of mutations in the CQ resistance transporter (PfCRT), which is located on the parasite digestive vacuole membrane (*Fidock et al., 2000*). Mutations in PfCRT are thought to mediate resistance by enabling drug efflux out of the digestive vacuole (*Wicht et al., 2020*; *Kim et al., 2019*; *Papakrivos et al., 2012*; *Sanchez et al., 2005*; *Summers et al., 2014*).

Heme is also central to the mechanism of action of artemisinins. Artemisinins contain a peroxide bridge that must be cleaved by heme for antimalarial activity (*Cumming et al., 1997*; *Meshnick et al., 1996*; *Kaiser et al., 2007*; *Zhang and Gerhard, 2008*; *Meunier and Robert, 2010*). Following activation, artemisinins nonspecifically alkylate adjacent molecules, causing widespread cellular damage to the parasite (*Wang et al., 2015*; *Ismail et al., 2016b*; *Jourdan et al., 2019*). Clinically, artemisinin resistance is primarily associated with mutations in Kelch 13 (K13) (*Ariey et al., 2014*; *Ashley et al., 2014*; *Straimer et al., 2015*), which are thought to reduce parasite hemoglobin uptake and digestion (*Birnbaum et al., 2020*; *Siddiqui et al., 2017*; *Yang et al., 2019*). Consequently, less heme is available to activate artemisinins.

ACTs remain the mainstay treatment for *falciparum* malaria. Widespread resistance to artemisinins in Southeast Asia and recent emergence of artemisinin resistance in Africa have posed great concern (*World Health Organization, 2024*; *Ashley et al., 2014*; *Uwimana et al., 2020*; *Uwimana et al., 2021*; *Tumwebaze et al., 2022*; *Rosenthal et al., 2024a*). Reduced artemisinin efficacy means that the partner drugs are responsible for clearing a larger parasite biomass, increasing the likelihood for resistance to emerge to partner drugs. Indeed, high treatment failure rates have been reported for AS-MFQ, DHA-PPQ, and AS-ADQ in Southeast Asia (*Saunders et al., 2014*; *Spring et al., 2015*; *Leang et al., 2015*; *van der Pluijm et al., 2019*; *van der Pluijm et al., 2020*; *Liu et al., 2023*; *Amaratunga et al., 2016*; *Phyo et al., 2016a*; *Mairet-Khedim et al., 2021*).

Given that heme is central to the mechanism of action of both quinolines and artemisinins, we hypothesized that these antimalarials would exhibit strong drug-drug interactions. Previously, several groups have assessed quinoline-artemisinin interactions using standard 48 hr or 72 hr assays and found that combinations were additive in CQ-sensitive parasites and additive to slightly antagonistic in CQ-resistant parasites (*Davis et al., 2006*; *Fivelman et al., 1999*; *Ribbiso et al., 2021*; *Ansbro et al., 2020*; *Muangnoicharoen et al., 2009*). However, standard 48 hr or 72 hr assays in which parasites are exposed to compounds for the entire assay duration do not well represent clinical drug exposure conditions and can mask not only resistance phenotypes, but also drug-drug interaction phenotypes (*Duru et al., 2015*; *Witkowski et al., 2013*; *Stokes et al., 2019*; *Gligorijevic et al., 2008*). We sought to re-evaluate quinoline-artemisinin interactions in multidrug-sensitive,

**Table 1.** Parasite strains used in this study.

Strains are annotated with their origin (isolate or edited), chloroquine resistance transporter (PfCRT) genotype, Kelch 13 (K13) genotype, and whether they are resistant to chloroquine (CQ), piperaquine (PPQ), or dihydroartemisinin (DHA). Wild-type (WT) K13 corresponds to the 3D7 genotype.

| Parasite name | Origin | PfCRT genotype | | | | | | | | | | K13 genotype | Resistance |
|---|---|---|---|---|---|---|---|---|---|---|---|---|---|
| | | 74 | 75 | 76 | 97 | 220 | 271 | 326 | 343 | 356 | 371 | | |
| 3D7 | Africa (1981) | M | N | K | H | A | Q | N | M | I | R | WT | – |
| Dd2 | Indochina (1980) | I | E | T | H | S | E | S | M | T | I | WT | CQ |
| Dd2 PfCRT[Dd2] | Edited (introns removed) | I | E | T | H | S | E | S | M | T | I | WT | CQ |
| Dd2 PfCRT[3D7] | Edited | M | N | K | H | A | Q | N | M | I | R | WT | – |
| Dd2 K13[R539T] | Edited | I | E | T | H | S | E | S | M | T | I | R539T | CQ, DHA |
| Dd2 PfCRT[M343L] | Edited | I | E | T | H | S | E | S | L | T | I | WT | CQ, PPQ |
| MRA-1284 (IPC_6261) | Cambodia (2012) | I | E | T | Y | S | E | S | M | T | I | C580Y | DHA, PPQ |

CQ-resistant, artemisinin-resistant, and PPQ-resistant parasites using pulsing assays that better represent artemisinin kinetics. Our work identifies that CQ, PPQ, and ADQ antagonize DHA, the active metabolite of clinically used artemisinins, likely by rendering heme chemically inert and preventing DHA activation. In addition, we explore drug-drug interactions beyond traditional ACTs, investigating interactions of putative triple ACTs (TACTs), as well as ozonide-quinoline combinations that failed in clinical trials. Overall, the data presented here highlight the importance of detailed investigation of drug-drug interactions, particularly in drug-resistant parasites.

## Results

### CQ and DHA are superantagonistic in CQ-resistant parasites

We began by evaluating CQ-DHA interactions in 3D7, a CQ-sensitive parasite, and Dd2, a CQ-resistant parasite (*Table 1*), using trophozoite-stage isobologram assays. Synchronized trophozoites were treated with fixed ratios of CQ and DHA for 4 hr, drugs were washed off, and then parasitemia was assessed 72 hr later. Given the short half-life of artemisinins (*Witkowski et al., 2013*), we believe that this pulsing format allows us to better assess the effect of ACT partner drugs on DHA activity compared to standard 48 hr or 72 hr assays. Fractional IC50 (FIC50) values were calculated from fixed ratios and then plotted on an XY graph to generate isobolograms. Isobolograms can be interpreted based on isobole shape and mean $\Sigma$ FIC50 values as follows: points along the dotted line with a mean $\Sigma$ FIC50 value close to 1 indicate additivity, points above the dotted line in a concave curve with a mean $\Sigma$ FIC50 value >1.25 indicate antagonism, and points below the dotted line in a convex curve with a mean $\Sigma$ FIC50 value <0.75 indicate synergy.

We observed strong antagonism for CQ-DHA in 3D7 parasites (*Figure 1a*), with a mean $\Sigma$ FIC50 value of 1.68. Antagonism was notably exacerbated in Dd2 parasites (*Figure 1b*), which had a mean $\Sigma$ FIC50 value of 2.44. The magnitude of this value far surpasses conventional antagonism, and we termed this phenomenon 'superantagonism'. Perhaps even more unexpected was the atypical shape of this isobole (*Figure 1b*), where points extended upward and peak $\Sigma$ FIC50 values exceeded 4. In contrast, peak $\Sigma$ FIC50 values for 3D7 reached only 2. The traditional isobole shape for 3D7 indicates a reciprocal relationship between CQ and DHA, whereas the atypical isobole shape of Dd2 suggests a one-sided relationship, whereby CQ is blocking DHA activity.

Multiple groups have demonstrated that parasite genetics can influence drug-drug interactions (*Ansbro et al., 2020*; *Rosenthal and Ng, 2021*; *Eastman et al., 2016*). 3D7 and Dd2 parasites differ genetically at a number of drug resistance markers, including PfCRT which is responsible for mediating CQ resistance. To determine if PfCRT genotype was responsible for differences in CQ-DHA interactions between these parasite lines, we evaluated interactions in two Dd2 parasites that are isogenic except at PfCRT (*Straimer et al., 2012*; *Ross et al., 2018*): Dd2 PfCRT[3D7], which is edited to have the 3D7 PfCRT genotype, and Dd2 PfCRT[Dd2], which maintains the Dd2 PfCRT genotype and was used as a cloning control (*Table 1*). Notably, superantagonism was lost in Dd2 PfCRT[3D7] parasites,

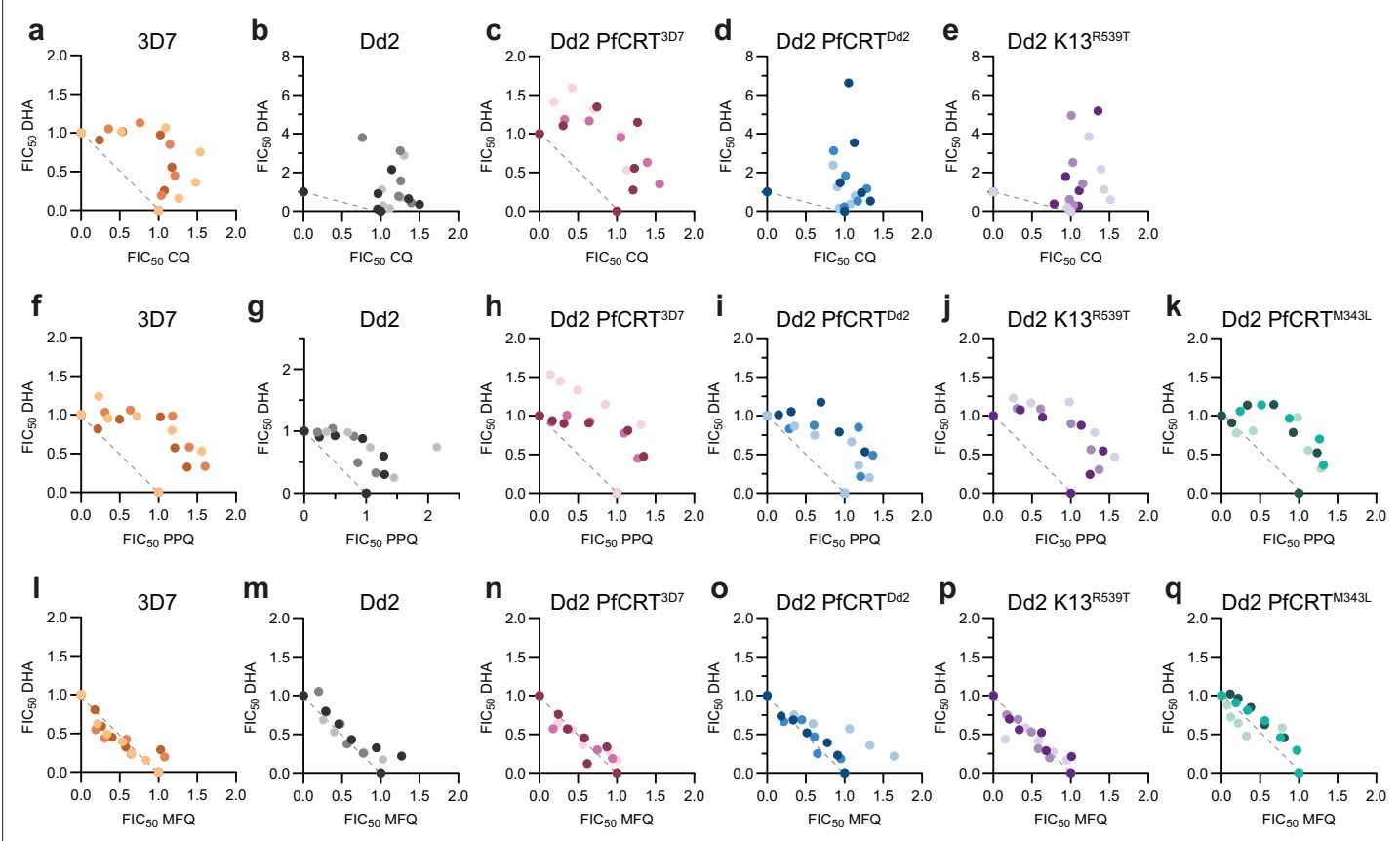

**Figure 1.** Chloroquine (CQ) and dihydroartemisinin (DHA) are superantagonistic in CQ-resistant parasites. Shown are trophozoite-stage isobolograms for (a–e) CQ-DHA, (f–k) PPQ-DHA, and (l–q) MFQ DHA. The dotted line on each graph represents perfect additivity. 3D7, Dd2, Dd2 PfCRT$^{3D7}$, Dd2 PfCRT$^{Dd2}$, Dd2 K13$^{R539T}$, and Dd2 PfCRT$^{M343L}$ parasites are indicated in orange, gray, pink, blue, purple, and teal, respectively. Data from three independent replicates are indicated by different shading.

The online version of this article includes the following source data and figure supplement(s) for figure 1:

**Source data 1.** Isobologram analysis.

**Figure supplement 1.** Chloroquine (CQ), dihydroartemisinin (DHA), piperaquine (PPQ), and mefloquine (MFQ) sensitivity of different parasites.

**Figure supplement 1—source data 1.** Chloroquine (CQ), dihydroartemisinin (DHA), piperaquine (PPQ), and mefloquine (MFQ) IC50 values.

which phenocopied 3D7 parasites, and had a mean $\Sigma$FIC50 value of 1.85 and a conventional isobole shape (*Figure 1c*). In contrast, Dd2 PfCRT$^{Dd2}$ parasites displayed one-sided superantagonism with a mean $\Sigma$FIC50 value of 2.73 and peak $\Sigma$FIC50 values >4.5 (*Figure 1d*). These data suggest that PfCRT genotype is the main genetic determinant that dictates CQ-DHA interactions.

Both 3D7 and Dd2 parasites are artemisinin sensitive, harboring a wild-type K13. To determine if resistance to artemisinins alters CQ-DHA interactions, we next evaluated CQ-DHA combinations in Dd2 parasites that are edited to have a K13$^{R539T}$ mutation (*Straimer et al., 2015*; *Table 1*). Dd2 K13$^{R539T}$ parasites, which also harbor a Dd2 PfCRT genotype, displayed a one-sided superantagonistic isobole that was reminiscent of both Dd2 and Dd2 PfCRT$^{Dd2}$ parasites (*Figure 1e*). This suggests that, unlike PfCRT genotype, K13 genotype appeared to have no impact on drug-drug interactions at the trophozoite stage.

## PPQ and DHA are antagonistic

Next, we sought to evaluate drug-drug interactions of DHA and the ACT partner drugs PPQ and MFQ. PPQ is believed to have a similar mechanism of action as CQ and both antimalarials are 4-aminoquinolines. Accordingly, we hypothesized that PPQ and DHA would also be antagonistic. In 3D7, Dd2, Dd2 PfCRT$^{3D7}$, Dd2 PfCRT$^{Dd2}$, and Dd2 K13$^{R539T}$ trophozoites, we observed conventional antagonism for PPQ-DHA (*Figure 1f–j*) with mean $\Sigma$FIC50 values ranging from 1.55 to 1.71.

To determine if, similar to CQ, PPQ-DHA interactions would be exacerbated in PPQ-resistant parasites, we also evaluated Dd2 PfCRT$^{M343L}$, which harbors the Dd2 PfCRT plus an additional mutation at M343L (*Ross et al., 2018*; *Table 1*). The M343L mutation had no impact on PPQ-DHA interactions and Dd2 PfCRT$^{M343L}$ parasites displayed conventional antagonism with a mean $\Sigma$FIC50 value of 1.58 (*Figure 1k*). Note that while the PPQ IC50 of Dd2 PfCRT$^{M343L}$ is 2.5-fold higher than Dd2 (65 nM versus 25 nM; *Figure 1—figure supplement 1*), this shift is substantially smaller than the >1000-fold CQ IC50 shift that we and others (*Gligorijevic et al., 2008*) have observed in CQ-sensitive versus CQ-resistant parasites using the pulsing assay format (*Figure 1—figure supplement 1*).

Next, we assessed interactions between MFQ and DHA (*Figure 1l–q*). Though MFQ contains a quinoline group, it is chemically classified as an aryl amino alcohol and is believed to have a heme-independent mechanism of action (*Vanaerschot et al., 2017*). Consistent with this, we observed additivity for MFQ-DHA in all six parasites tested with mean $\Sigma$FIC50 values ranging from 0.94 to 1.20. This additivity, along with the synergism that we previously reported for DHA-proteasome inhibitor combinations (*Rosenthal and Ng, 2021*), provides context for the antagonism and superantagonism that we observed for PPQ-DHA and CQ-DHA.

## CQ combination treatment confers artemisinin resistance to K13$^{WT}$ parasites in ring-stage survival assays

Given our observation that CQ and PPQ antagonize DHA in trophozoite stages, we next sought to determine if these quinolines would promote survival in early ring stages where artemisinin resistance is classically observed. We performed DHA dose-response assays on early ring-stage parasites to determine both ring-stage survival assay (RSA) values and IC50 values (*Figure 2a* and *Figure 2—figure supplement 1*). The RSA is used to delineate artemisinin-sensitive versus resistant parasites, where survival >1% corresponds to resistance and <1% indicates sensitivity (*Witkowski et al., 2013*). Though DHA IC50 values cannot reliably differentiate artemisinin-sensitive versus resistant parasites (*Witkowski et al., 2013*), they can provide important insight into different factors that contribute to artemisinin susceptibility (*Wang et al., 2016*).

In the absence of quinolines, the artemisinin-sensitive K13$^{WT}$ parasites, Dd2 PfCRT$^{3D7}$ and Dd2, had RSA survival values ≤1%, while Dd2 K13$^{R539T}$ had an RSA survival of 12.63% (*Figure 2b–d*, *top panel*). Co-treatment with 10 µM CQ, a pharmacologically relevant concentration (*Cabrera et al., 2009*; *Melo et al., 2022*), significantly increased RSA survival, conferring artemisinin resistance to both Dd2 PfCRT$^{3D7}$ and Dd2 parasites with mean RSA values of 2.66% and 4.49%, respectively (*Figure 2b and c*, *top panel*). RSA survival values were also increased in Dd2 K13$^{R539T}$ parasites to 18.39%, although this shift was not statistically significant (*Figure 2d*, *top panel*). When assessing DHA sensitivity by ring-stage IC50 values, CQ co-treatment also had a substantial rescuing effect in both Dd2 and Dd2 K13$^{R539T}$ parasites, resulting in a fourfold increase in DHA IC50 (*Figure 2c and d*, *bottom panel*). Note that while we were unable to determine a DHA IC50 value for CQ-sensitive Dd2 PfCRT$^{3D7}$ parasites in the presence of 10 µM CQ, the dose-response curve was suggestive of antagonism (*Figure 2—figure supplement 1a*).

To determine if PPQ would have a similar rescuing effect, we performed these same assays in the presence of 200 nM PPQ. This concentration was chosen because it is sublethal for early rings in this short 3 hr pulse, pharmacologically relevant (*Nguyen et al., 2009*; *Hoglund et al., 2017*), and mimics the PPQ survival assay (PSA) that is used to delineate PPQ resistance (*Duru et al., 2015*). Co-treatment with 200 nM PPQ did not have a rescuing effect for Dd2 PfCRT$^{3D7}$, Dd2, or Dd2 K13$^{R539T}$ parasites and resulted in mean RSA values that were either similar to or lower than the DHA-only control (*Figure 2b–d*). Note that in the absence of DHA, 200 nM PPQ inhibited parasite growth approximately 20%. However, co-treatment with 200 nM PPQ did not shift the IC50 values or DHA dose-response curves of these parasites (*Figure 2b–d* and *Figure 2—figure supplement 1a–c*), indicating that PPQ and DHA are also antagonistic in early ring stages.

We were curious if higher concentrations of PPQ might have a rescuing effect, particularly for PPQ-resistant parasites that are capable of surviving a higher PPQ dose. To this end, we assayed two PPQ-resistant parasites: Dd2 PfCRT$^{M343L}$ and MRA-1284 (IPC_6261), which is an artemisinin-resistant and PPQ-resistant clinical isolate that was collected in 2012 from Cambodia (*Table 1*). Dd2 PfCRT$^{M343L}$ and MRA-1284 have PPQ survival values (PSA) of 10% (*Ross et al., 2018*) and 40% (*Nkhoma et al., 2021*), respectively. While co-treatment with 10 µM CQ had a rescuing effect for Dd2 PfCRT$^{M343L}$ as assessed

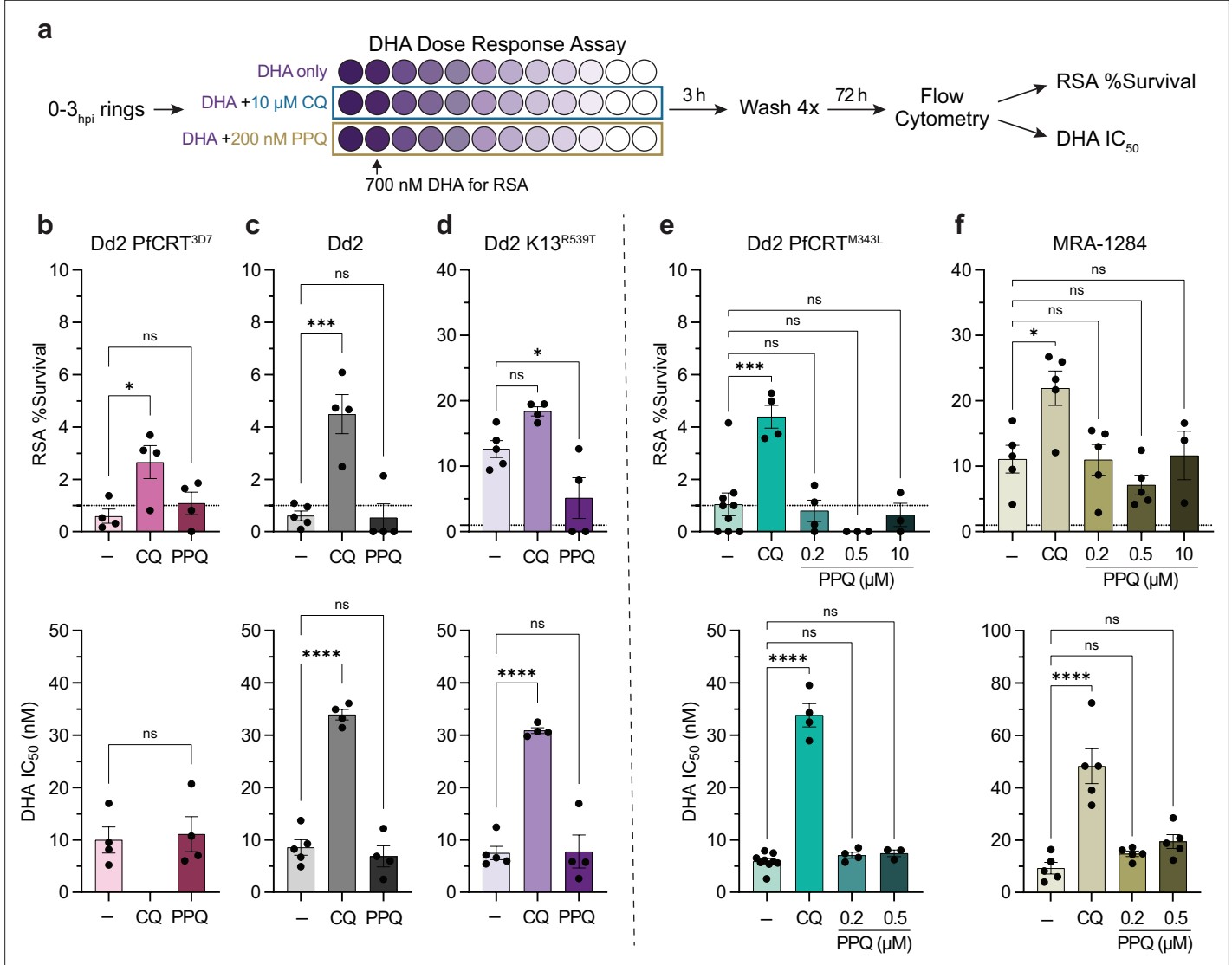

**Figure 2.** Chloroquine (CQ) co-treatment confers artemisinin resistance in K13[WT] parasites. (**a**) Dihydroartemisinin (DHA) dose-response assays were performed on 0–3 hr post invasion (hpi) ring-stage parasites. Twofold serial dilutions of DHA were prepared in 96-well plates (DHA concentration represented by different shades of purple) starting at 1.4 µM so that both IC50 values and ring-stage survival assay (RSA) survival values at 700 nM DHA could be determined. To evaluate quinoline-DHA interactions in early ring stages, DHA titrations were prepared with fixed concentrations of 10 µM CQ or 200 nM piperaquine (PPQ). (**b–d**) Mean RSA survival values ± SEM (*top panel*) and mean DHA IC50 values ± SEM (*bottom panel*) are shown for (**b**) Dd2 PfCRT[3D7], (**c**) Dd2, and (**d**) Dd2 K13[R539T] parasites. For the PPQ-resistant parasites (**e**) Dd2 PfCRT[M343L] and (**f**) MRA-1284, additional fixed PPQ concentrations of 500 nM and 10 µM were also tested. At least three independent replicates were performed for each experiment. For b *bottom panel*, a Student's t-test was used to assess statistical differences between DHA alone versus DHA + PPQ. For all other graphs, statistical differences between DHA alone (-) and DHA + quinolines was assessed using a one-way ANOVA with a Dunnett's test for multiple comparisons. ****$p < 0.0001$; ***$p < 0.001$; *$p < 0.05$; ns = not significant.

The online version of this article includes the following source data and figure supplement(s) for figure 2:

**Source data 1.** Ring-stage survival assay (RSA) dose-response assays.

**Figure supplement 1.** Early ring-stage dose-response curves.

by RSA and IC50 values, neither 200 nM PPQ, 500 nM PPQ, nor 10 µM PPQ co-treatment rescued parasite survival (***Figure 2e*** and ***Figure 2—figure supplement 1d***). Similarly, 10 µM CQ co-treatment decreased MRA-1284 sensitivity to DHA as assessed by RSA and early ring IC50 values, while co-treatment with 200 nM PPQ or 500 nM PPQ had no effect on DHA sensitivity (***Figure 2f***). This phenotype is consistent with conventional antagonism. Interestingly, while co-treatment with 10 µM PPQ did not alter the RSA survival of MRA-1284, it did shift the dose-response curve to the left (***Figure 2—figure***

*supplement 1e*). However, since 10 µM PPQ inhibited parasite growth approximately 60% on its own, we were unable to determine an IC50 value.

## CQ and PPQ interfere with heme iron reactivity

In total, we profiled three different quinoline-DHA interactions, which are summarized in *Figure 3a*. In solution, quinolines and heme form heme-quinoline complexes, wherein heme is present as a µ-oxo dimer (*Kuter et al., 2016*; *Dorn et al., 1998*; *Leed et al., 2002*). Though the exact stoichiometry of this interaction for each quinoline remains debated (*Kuter et al., 2016*; *Dorn et al., 1998*; *Leed et al., 2002*; *Egan, 2006*; *Vippagunta et al., 2000*), formation of such a complex could render quinoline-bound heme chemically inert (i.e. unable to activate artemisinins).

To investigate this, we utilized a recently developed small molecule probe, H-FluNox, that is highly sensitive and specific for heme over hemin, heme-bound proteins, $Fe^{2+}$, and other salts and metal ions (*Kawai et al., 2022*). H-FluNox is basally non-fluorescent, but upon cleavage of an N-O bond by the $Fe^{2+}$ of heme, H-FluNox fluoresces (*Kawai et al., 2022*; *Figure 3b*, *top panel*). While not identical, this reaction is similar to artemisinin activation, which requires cleavage of artemisinin's peroxide bond by the $Fe^{2+}$ of heme (*Meunier and Robert, 2010*; *Figure 3b*, *bottom panel*). To determine if quinoline-heme complexes would react with H-FluNox, we began by assessing H-FluNox fluorescence in solution with 10 µM CQ, which is sub-lethal for CQ-resistant parasites and pharmacologically relevant (*Cabrera et al., 2009*; *Melo et al., 2022*). Relative to the untreated control, 10 µM CQ inhibited fluorescence approximately 50% in the presence of 250 nM heme (*Figure 3c*). This suggests that heme iron reactivity is impaired when heme is in complex with CQ. Next, we evaluated PPQ and MFQ at two different concentrations: 150 nM, which reflects a 5× trophozoite-stage IC50 concentration, and 10 µM to directly compare these quinolines with CQ. At 150 nM, both PPQ and MFQ inhibited fluorescence, albeit only 10% (*Figure 3c*). At 10 µM, MFQ inhibited fluorescence to a similar extent as CQ, while 10 µM PPQ only inhibited fluorescence 25% (*Figure 3c*). Note that 10 µM concentrations exceed maximum plasma concentrations of PPQ and MFQ, which have been reported to reach approximately 0.5–1 µM and 5 µM, respectively (*Nguyen et al., 2009*; *Hoglund et al., 2017*; *Gutman et al., 2009*).

Next, we assessed fluorescence in live trophozoite-stage parasites using the cell-permeable analog of H-FluNox, Ac-H-FluNox (*Kawai et al., 2022*). In untreated Dd2 parasites, fluorescence was predominantly observed in the parasite cytoplasm (*Figure 3d*), and little signal was detected in red blood cells or the parasite digestive vacuole. This suggests that a significant portion of DHA could be activated in the parasite cytoplasm. Lack of signal in the digestive vacuole could be due to issues with permeability, the specificity of Ac-H-FluNox for heme over hemin (the predominant heme species in the digestive vacuole), or sensitivity of Ac-H-FluNox to a low pH environment. In solution, H-FluNox fluorescence was decreased approximately 60% at pH 5.4 (*Figure 3—figure supplement 1a*), which is the estimated pH of the digestive vacuole (*Wiser, 2024*). However, alkalinization of the digestive vacuole with ammonium chloride (*Yayon et al., 1984*) did not increase digestive vacuole fluorescence signal (*Figure 3—figure supplement 1b*), suggesting that low pH alone cannot explain this lack of labeling.

Importantly, treatment of Dd2 trophozoites with 10 µM CQ nearly ablated Ac-H-FluNox fluorescent signal (*Figure 3d and e*). Treatment with 150 nM PPQ also significantly reduced fluorescence, though not to the same extent as 10 µM CQ (50% inhibition versus >92% inhibition) (*Figure 3d and e*). In contrast, parasites treated with 150 nM MFQ had similar fluorescence as untreated control parasites (*Figure 3d and e*). These assays with H-FluNox and Ac-H-FluNox correlate well with hemozoin inhibition assays in which CQ, PPQ, and MFQ inhibit hemozoin formation in solution (*Dorn et al., 1998*), but only CQ and PPQ inhibit hemozoin formation in parasites (*Combrinck et al., 2013*; *Dhingra et al., 2017*; *Warhurst et al., 2007*; *Combrinck et al., 2015*). To determine if concentration alone was responsible for the differences that we observed between CQ and the other quinolines, we also evaluated Ac-H-FluNox fluorescence following treatment with 10 µM PPQ or 10 µM MFQ. At this super-physiological and super-lethal concentration, an equimolar concentration of CQ still inhibited H-FluNox fluorescence to a greater extent than PPQ or MFQ (*Figure 3—figure supplement 2*). This suggests that something unique to the chemistry of CQ and/or the biology of the parasite is responsible for the differences that we observed.

PfCRT^Dd2 is believed to mediate CQ resistance by pumping CQ out of the digestive vacuole (*Wicht et al., 2020*; *Kim et al., 2019*; *Papakrivos et al., 2012*; *Sanchez et al., 2005*; *Summers et al., 2014*),

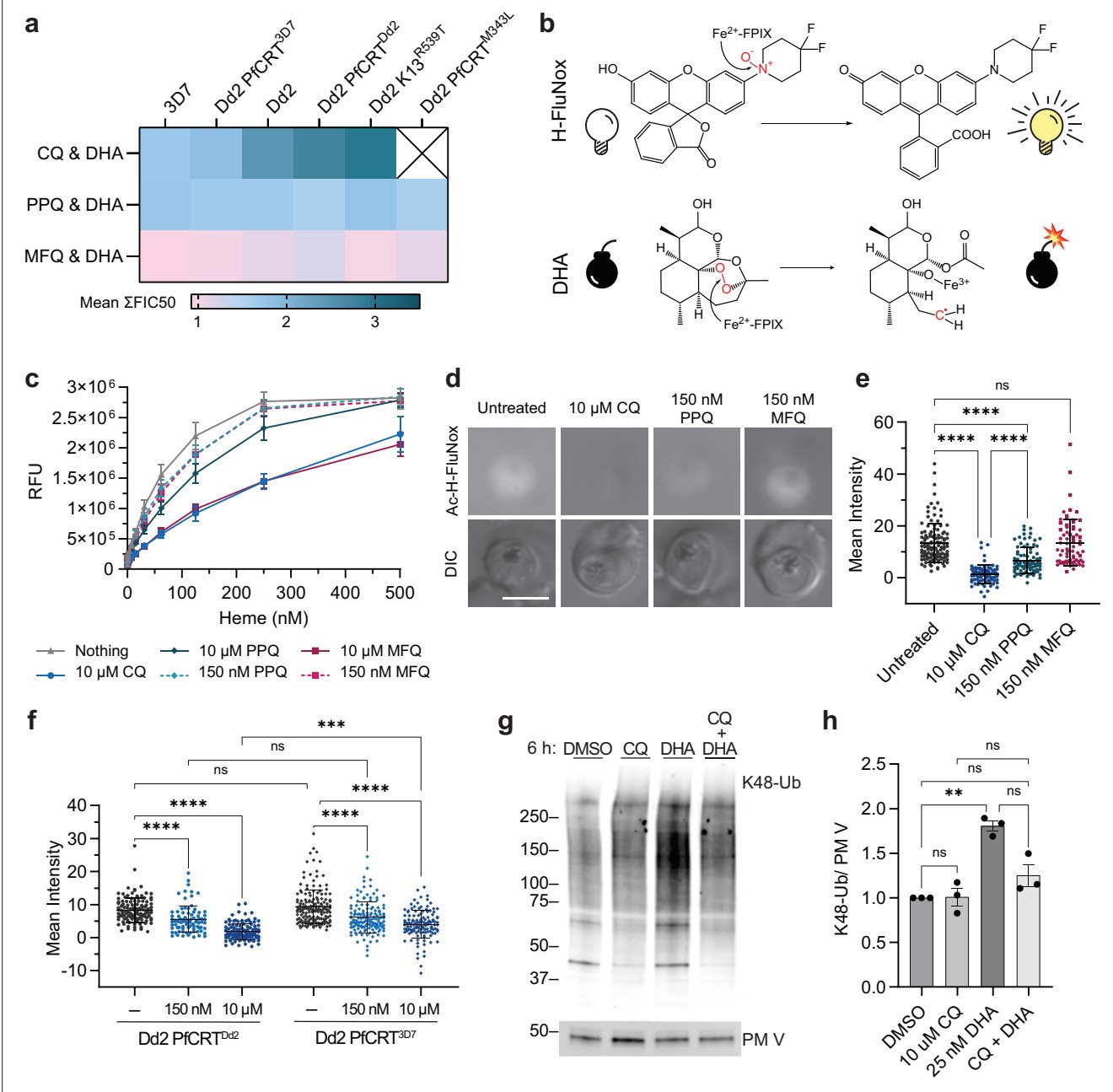

**Figure 3.** Chloroquine (CQ) and piperaquine (PPQ) antagonize dihydroartemisinin (DHA) activation by 'inactivating' heme. (**a**) Shown are mean $\Sigma$ FIC50 values calculated from the trophozoite-stage isobolograms in **Figure 1**. Values close to 1 (pink) indicate additivity. Values ranging from 1.25 to 2 (light blue) indicate classical antagonism. Values >2.25 (dark turquoise) indicate superantagonism. (**b** *top panel*) Heme (Fe2+-FPIX) cleaves the N-O bond of H-FluNox, mediating probe fluorescence. (**b** *bottom panel*) Heme cleaves the peroxide bridge of DHA, which generates a carbon-centered radical that can alkylate proximal biomolecules. (**c**) H-FluNox was incubated with increasing concentrations of heme in the presence or absence of 10 μM CQ, 150 nM PPQ, 10 μM PPQ, 150 nM MFQ, or 10 μM MFQ. Shown are mean relative fluorescence units (RFU) ± SD from at least three independent replicates performed in technical triplicate. (**d** and **e**) Dd2 trophozoites were treated with 10 μM CQ, 150 nM PPQ, 150 nM MFQ, or mock-treated for 5.5 hr. (**f**) Dd2 PfCRT^Dd2 and Dd2 PfCRT^3D7 trophozoites were mock-treated or treated with 150 nM CQ or 10 μM CQ for 5.5 hr. (**d–f**) Parasites were then incubated with 10 μM Ac-H-FluNox (the cell-permeable analog of H-FluNox) to visualize and quantify 'active' heme. Shown are (**d**) representative images and (**e** and **f**) mean fluorescence intensity ± SD of at least 70 parasites from at least three independent drug treatments. Statistical significance was assessed using a one-way ANOVA with a Tukey's test for multiple comparisons. (**g** and **h**) Dd2 trophozoites were treated with 10 μM CQ, 25 nM DHA, 10 μM CQ + 25 nM DHA for 6 hr. Parasite lysates were subjected to western blot and probed with anti-K48-linked ubiquitin (K48-Ub) antibodies or anti-plasmepsin V (PMV) antibodies. Shown is a (**g**) representative blot and (**h**) quantification of relative K48-Ub intensity ± SEM from three independent

*Figure 3 continued on next page*

*Figure 3 continued*

replicates. Statistical significance between mock-treated and antimalarial-treated parasites was determined using a one-way ANOVA with a Tukey's test for multiple comparisons. ****$p<0.0001$; ***$p<0.001$; **$p<0.01$; ns = not significant. Scale bar = 5 µm.

The online version of this article includes the following source data and figure supplement(s) for figure 3:

**Source data 1.** Mean sum FIC50 values used to generate the heat map.

**Source data 2.** H-FluNox relative fluorescence units (RFU) with chloroquine (CQ), piperaquine (PPQ), and mefloquine (MFQ).

**Source data 3.** Ac-H-FluNox parasite fluorescence with chloroquine (CQ), piperaquine (PPQ), and mefloquine (MFQ).

**Source data 4.** Ac-H-FluNox fluorescence of Dd2 PfCRT$^{Dd2}$ and Dd2 PfCRT$^{3D7}$ parasites.

**Source data 5.** Uncropped, labeled western blots.

**Source data 6.** Western blot raw images.

**Source data 7.** Western blot quantifications.

**Figure supplement 1.** Effect of pH on H-FluNox and Ac-H-FluNox activity.

**Figure supplement 1—source data 1.** H-FluNox pH titration relative fluorescence units (RFU).

**Figure supplement 2.** Equimolar comparison of chloroquine (CQ), piperaquine (PPQ), and mefloquine (MFQ) on Ac-HFluNox fluorescence.

**Figure supplement 2—source data 1.** Ac-H-FluNox fluorescence for 10 µM chloroquine (CQ), 10 µM piperaquine (PPQ), 10 µM mefloquine (MFQ) treatment.

**Figure supplement 3.** Representative images of Dd2 PfCRT$^{Dd2}$ and Dd2 PfCRT$^{3D7}$ Ac-H-FluNox fluorescence.

suggesting that cytoplasmic CQ concentrations may differ between PfCRT$^{Dd2}$ and PfCRT$^{3D7}$ parasites treated with the same concentration of CQ. Given the substantial difference that we saw in the shape of CQ-DHA isobolograms between PfCRT$^{Dd2}$ and PfCRT$^{3D7}$ parasites, we were curious if differences in PfCRT genotype would impact Ac-H-FluNox fluorescence. Accordingly, we treated Dd2 PfCRT$^{Dd2}$ and Dd2 PfCRT$^{3D7}$ parasites with 150 nM CQ or 10 µM CQ and measured chemically 'reactive' heme using Ac-H-FluNox. In untreated Dd2 PfCRT$^{Dd2}$ versus Dd2 PfCRT$^{3D7}$ parasites, we observed no difference in Ac-H-FluNox localization or fluorescence intensity (*Figure 3f* and *Figure 3—figure supplement 3*). Following treatment with 150 nM CQ, fluorescence was inhibited by 25% in Dd2 PfCRT$^{3D7}$ and by 35% in Dd2 PfCRT$^{Dd2}$; however, this difference was not statistically significant (*Figure 3f*). Curiously, while 10 µM CQ treatment reduced mean fluorescence approximately 60% in Dd2 PfCRT$^{3D7}$ parasites, it was not to the extent of Dd2 PfCRT$^{Dd2}$ parasites (80%) (*Figure 3f*). These data suggest that transport of CQ out of the digestive vacuole by PfCRT$^{Dd2}$ may contribute to superantagonism between CQ and DHA in CQ-resistant parasites.

## CQ co-treatment rescues DHA-mediated protein damage

Following activation by heme, DHA causes widespread protein damage in the parasite (*Wang et al., 2015*; *Ismail et al., 2016b*; *Jourdan et al., 2019*; *Bridgford et al., 2018*). Accordingly, we would expect that if CQ is antagonizing DHA activation, then CQ co-treatment should rescue DHA-mediated protein damage. To test this hypothesis, we treated Dd2 trophozoites with CQ, DHA, or CQ+DHA and probed for K48-linked ubiquitin, which is a marker of protein damage (*Bridgford et al., 2018*). Consistent with what has been previously reported (*Rosenthal et al., 2024b*), CQ treatment alone had no effect on K48-linked ubiquitin levels, while treatment with DHA resulted in a twofold increase in K48-linked ubiquitin (*Figure 3g and h*). Importantly, the addition of CQ to DHA had a rescuing effect, resulting in levels of K48-linked ubiquitin that were similar to the DMSO-treated control (*Figure 3g and h*). Collectively, these data suggest that CQ antagonizes DHA by rendering heme chemically inert, reducing DHA activation, and consequently, reducing DHA-mediated protein damage.

## PPQ and MFQ do not harm treatment efficacy but strongly antagonize each other

With the looming threat of widespread ACT treatment failure, several ideas have been proposed to delay the spread of ACT resistance by using currently available antimalarials. TACTs, which are composed of a traditional ACT plus an additional longer-lived partner drug, have gained the most traction. Early clinical trials indicate that DHA-PPQ-MFQ and AM-LM-ADQ show good safety and efficacy against artemisinin-sensitive and resistant parasites (*van der Pluijm et al., 2020*; *Coulibaly*

*et al., 2015*; *Peto et al., 2022*; *Mahamar et al., 2025*). We were curious how the addition of a third antimalarial would influence drug interactions. Accordingly, we performed trophozoite-stage DHA dose-response assays on Dd2 parasites in the presence of a fixed, sub-lethal concentration of PPQ, MFQ, or PPQ+MFQ. Addition of 2 nM DHA was also included as a control to represent additivity. As assessed by DHA IC50 values, addition of 15 nM PPQ did not improve parasite killing, while addition of 15 nM MFQ resulted in a threefold decrease in IC50 compared to the DMSO control (*Figure 4a and b*). These phenotypes are consistent with antagonistic and additive relationships, respectively. Notably, the addition of PPQ+MFQ had no additional benefit compared to the addition of MFQ alone (*Figure 4a and b*).

Following the initial 3-day TACT treatment regimen, long-lived partner drugs are responsible for clearing the remaining parasite burden. Accordingly, we sought to assess drug-drug interactions of PPQ-MFQ in the absence of DHA. Previously, it was reported that MFQ and PPQ were moderately antagonistic in 3D7 and K1 parasites using a standard 48 hr assay, with mean $\Sigma$ FIC50 values of 1.31 and 1.28, respectively (*Davis et al., 2006*). Using our trophozoite-stage assay, we observed reciprocal superantagonism in both PPQ-sensitive Dd2 (*Figure 4c*) and PPQ-resistant Dd2 PfCRT[M343L] (*Figure 4—figure supplement 1a*), with mean $\Sigma$ FIC50 values of 2.62 and 2.44, respectively. This phenotype appears to be unique to PPQ, as we observed only strong antagonism for CQ-MFQ with a mean $\Sigma$ FIC50 of 1.72 (*Figure 4—figure supplement 1b*). Overall, these in vitro data suggest that based on drug-drug interactions, PPQ and MFQ may not be a strategic long-lived antimalarial pairing for TACTs.

## Quinolines antagonize peroxides

To determine if the drug-drug interactions that we evaluated were specific to a subset of antimalarials or represented a broader phenotype, we next evaluated DHA interactions with the clinically used ACT partner drugs ADQ and LM. Structurally, ADQ is classified as a 4-aminoquinoline like PPQ and CQ, while LM is classified as an aryl amino alcohol like MFQ. However, unlike MFQ, LM lacks a quinoline group. We found that in Dd2 parasites, ADQ and DHA were antagonistic (*Figure 4d*) with a mean $\Sigma$ FIC50 of 1.61, while LM and DHA were additive (*Figure 4e*) with a mean $\Sigma$ FIC50 of 1.02.

To determine if 4-aminoquinolines antagonize other peroxide antimalarials, we next sought to assess interactions of two different OZ439-quinoline pairs that were previously evaluated in clinical trials. Similar to artemisinins, the synthetic ozonide, OZ439, requires activation via heme-mediated cleavage of a peroxide bond but has a significantly improved half-life of 46–62 hr (*Phyo et al., 2016b*). In phase II clinical trials, OZ439 showed good efficacy alone (*Phyo et al., 2016b*), but poor efficacy either in combination with PPQ (*Macintyre et al., 2017*) or the next-generation quinoline, ferroquine (FQ) (*Adoke et al., 2021*; *Gansane et al., 2023*). To assess drug-drug interactions, we performed trophozoite-stage isobolograms with OZ439-PPQ and OZ439-FQ in Dd2 parasites. The pulsing format was used to maintain consistency across experiments. Both PPQ-OZ439 (*Figure 4f*) and FQ-OZ439 (*Figure 4g*) were antagonistic with mean $\Sigma$ FIC50 values of 1.42 and 1.52, respectively. These data suggest that antagonistic interactions are potentially one of several factors (*Macintyre et al., 2017*; *Adoke et al., 2021*; *Gansane et al., 2023*; *Hamilton et al., 2019*; *Phuc et al., 2017*) that may have contributed to the poor efficacy of these single-dose combination therapies.

Although the mechanism of action of ADQ and FQ is less well studied, both antimalarials appear to inhibit hemozoin formation. ADQ was shown to inhibit hemozoin formation in parasites using the heme fractionation assay (*Combrinck et al., 2015*). While in-parasite assays have not been performed with FQ, in vitro biochemical studies suggest that FQ may inhibit hemozoin formation even better than CQ (*Biot et al., 2005*). Accordingly, we were curious if ADQ and FQ treatment would 'inactivate' heme in parasites. Using the Ac-H-FluNox heme probe, we observed that relative to the untreated control, treatment with 150 nM ADQ or 150 nM FQ (approximately 5× IC50 concentrations; *Figure 4—figure supplement 2*) reduced Ac-H-FluNox fluorescent signal approximately 50% and 55%, respectively (*Figure 4h and i*). In contrast, treatment with an equipotent concentration of LM had no significant impact on Ac-H-FluNox signal relative to the untreated control (*Figure 4h and i*). Among the six partner drugs that were evaluated, we observed a clear inverse correlation between Ac-H-FluNox signal and mean $\Sigma$ FIC50, with an $R^2$ value of 0.84 (*Figure 4j*). These data strongly suggest that antimalarials that render heme inert also antagonize artemisinin and ozonide antimalarials, presumably by blocking endoperoxide activation.

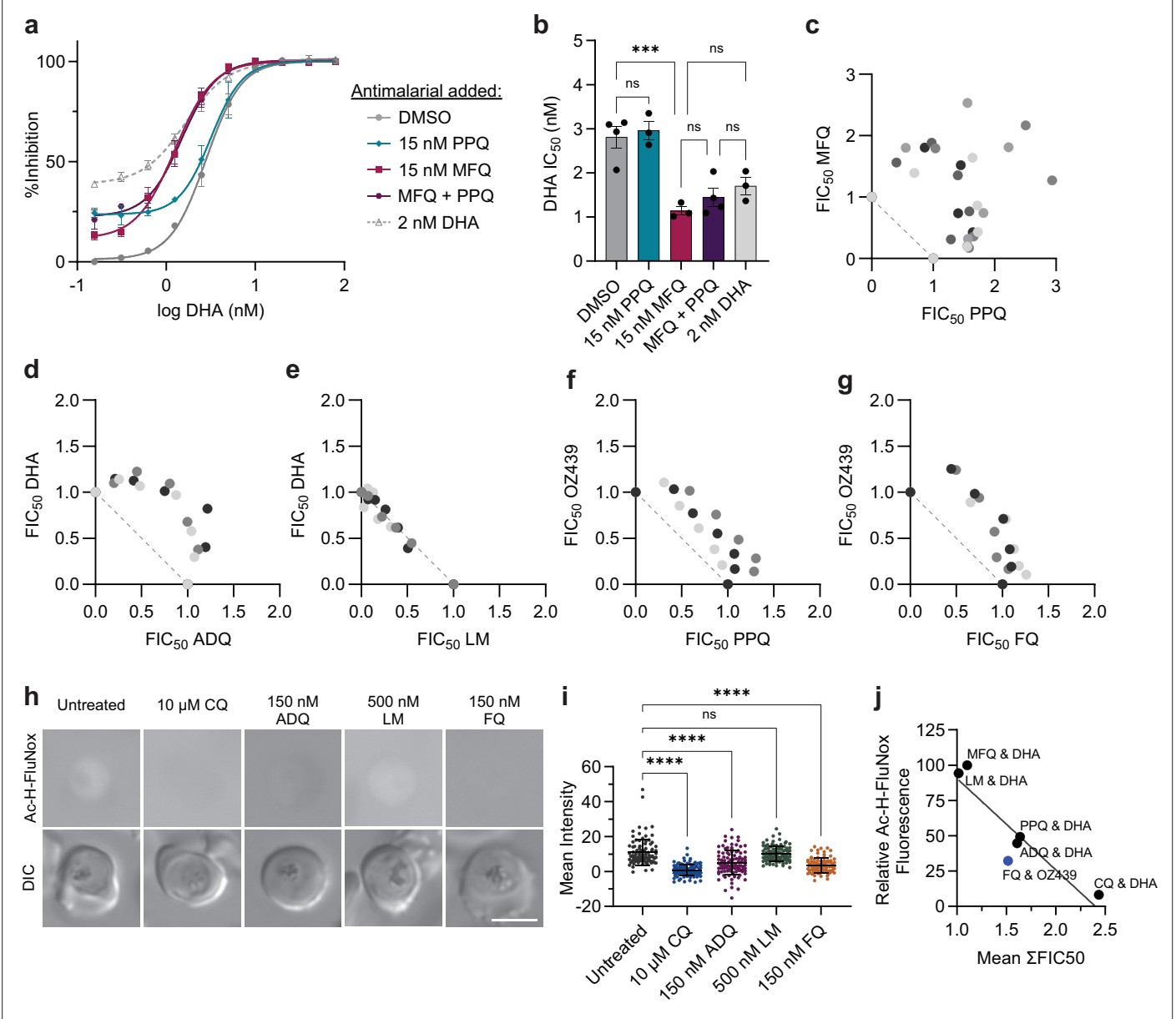

**Figure 4.** Quinolines antagonize peroxides. (**a** and **b**) Trophozoite-stage dihydroartemisinin (DHA) dose-response assays were performed with Dd2 parasites in the presence or absence of 15 nM piperaquine (PPQ), 15 nM mefloquine (MFQ), 15 nM PPQ+15 nM MFQ, or 2 nM DHA. Shown are (**a**) dose-response curves with mean % inhibition ± SEM and (**b**) mean IC50 values ± SEM from at least three independent replicates. Statistical significance was determined using a one-way ANOVA with a Tukey's test for multiple comparisons. Trophozoite-stage isobolograms were performed on Dd2 parasites to determine drug-drug interactions of (**c**) PPQ-MFQ, (**d**) ADQ-DHA, (**e**) LM-DHA, (**f**) PPQ-OZ439, and (**g**) FQ OZ439 using the following fixed ratios: 1:0, 4:1, 2:1, 1:1, 1:2, 1:4, 0:1. Shown are fractional IC50 values from at least three independent replicates. Independent replicates are indicated in different shading and the diagonal dotted line on each plot indicates perfect additivity. (**h** and **i**) Dd2 parasites were mock-treated or treated with 10 μM chloroquine (CQ), 150 nM ADQ, 500 nM lumefantrine (LM), or 150 nM ferroquine (FQ) for 5.5 hr and free heme was then labeled in live parasites with Ac-H-FluNox. Shown are (**h**) representative images and (**i**) mean fluorescence intensity ± SD of at least 90 parasites from three independent drug treatments. Statistical significance was assessed using a one-way ANOVA with a Dunnett's test for multiple comparisons. ****p<0.0001; ns = not significant. (**j**) Mean relative fluorescence of Ac-H-FluNox from drug treatments in *Figure 3* and (**i**) was plotted against Dd2 mean ΣFIC50 values for the indicated combinations. Combinations with DHA are indicated in black. Combinations with OZ439 are indicated in blue. Scale bar = 5 μm.

The online version of this article includes the following source data and figure supplement(s) for figure 4:

**Source data 1.** Dose response analyses for TACTs.

**Source data 2.** Isobologram analyses.

*Figure 4 continued on next page*

*Figure 4 continued*

**Source data 3.** Ac-H-FluNox fluorescence for ADQ, FQ, and LM.

**Source data 4.** Values for FIC50 and Ac-H-FluNox correlation.

**Figure supplement 1.** Additional mefloquine (MFQ) isobolograms.

**Figure supplement 1—source data 1.** Isobologram analysis.

**Figure supplement 2.** Amodiaquine (ADQ), lumefantrine (LM), and ferroquine (FQ) IC50 values.

**Figure supplement 2—source data 1.** Amodiaquine (ADQ), lumefantrine (LM), ferroquine (FQ), IC50 values.

## Discussion

Combination therapies are widely believed to delay the emergence of drug resistance not only for malaria, but also bacterial infections, HIV, fungal infections, and cancer. However, to maximize the benefits of combination therapies, it is essential that drugs are strategically paired. At a minimum, combinations should enhance efficacy to reduce parasite biomass and reduce the potential for drug resistance to evolve. In vitro assessment of drug-drug interactions between two antimalarials with vastly different half-lives is challenging. We chose to use a short treatment pulse to mimic artemisinin exposure and gain an understanding of how quinolines impact artemisinin activity. While drug-drug interactions are undoubtedly much more complex in vivo, we believe this assay format better recapitulates clinical conditions compared to traditional 48 hr or 72 hr assays.

We found that CQ and PPQ are antagonistic with DHA as assessed in early ring and trophozoite stages across three different parasite backgrounds. Artemisinin resistance is thought to be mediated by reduced heme-mediated activation and/or enhanced parasite stress response. Accordingly, 4-aminoquinolines could antagonize DHA by either or both mechanism(s). Our work with Ac-H-FluNox suggests that 4-aminoquinolines antagonize DHA by rendering intra-parasitic heme chemically inert, thus blocking DHA activation. However, future work will be needed to assess the effect of 4-aminoquinolines on parasite stress responses.

To date, it remains debated where artemisinins are activated within the parasite. There are several lines of evidence that suggest that artemisinins can be activated in the parasite cytoplasm. First, immunofluorescence studies with an anti-AS antibody (*Wang et al., 2010*) and live-cell imaging with an artemisinin-based photoaffinity probe (*Gao et al., 2024*) revealed that artemisinins localize throughout the parasite cytoplasm. Second, numerous chemical proteomics studies utilizing artemisinin and ozonide-based click chemistry probes have identified a number of cytoplasmic proteins as targets of artemisinin and ozonides (*Wang et al., 2015*; *Ismail et al., 2016b*; *Jourdan et al., 2019*; *Ismail et al., 2016a*; *Siddiqui et al., 2022*). Given how quickly activated artemisinins react with adjacent molecules, it is unlikely that activated artemisinins could traverse the digestive vacuole or other organelle membranes before encountering a cytoplasmic target. Finally, immunofluorescence assays with antibodies specific for the alkylation signature of the ozonide OZ277 (*Jourdan et al., 2016*) revealed labeling throughout the parasite cytoplasm. Our data with Ac-H-FluNox also lend credence to this hypothesis. We observed Ac-H-FluNox fluorescence throughout the parasite cytoplasm, indicating that there is a substantial amount of heme in this location that is chemically competent to activate artemisinins. It is possible that technical limitations, such as issues with membrane permeability, limit Ac-H-FluNox fluorescence in the digestive vacuole. Nevertheless, these data collectively indicate that artemisinins can be activated in the parasite cytoplasm.

Previously, it has been reported that CQ treatment increases intracellular 'free' heme (*Combrinck et al., 2013*; *Abshire et al., 2017*; *Famin and Ginsburg, 2003*). However, the term 'free' heme is somewhat malleable. While the pyridine-heme fractionation assay (*Combrinck et al., 2013*; *Combrinck et al., 2015*), electron spectroscopic imaging with energy-loss spectroscopy (*Combrinck et al., 2013*), and heme biosensor (*Abshire et al., 2017*) can be used to detect and quantify heme species, they are unable to differentiate chemically 'active' heme from inert heme (i.e. quinoline bound or protein bound). In contrast, H-FluNox is highly specific for chemically 'active' heme (*Kawai et al., 2022*). One possibility to reconcile our data with previous reports is that CQ treatment increases non-hemozoin/hemoglobin heme, but that this heme is bound to CQ or other biomolecules, rendering it unable to activate Ac-H-FluNox or peroxides. It is important to note that the chemistry of artemisinin and Ac-H-FluNox activation is not identical, and these data with Ac-H-FluNox should be interpreted with

caution. Nevertheless, we believe that Ac-H-FluNox is the best currently available proxy to assess artemisinin activation in live parasites.

CQ-DHA antagonism was notably exacerbated in CQ-resistant parasites. The concentrations used in isobolograms are chosen based on parasite IC50 values to individual compounds. While all parasites were exposed to similar DHA concentrations, CQ-resistant parasites were exposed to much higher CQ concentrations compared to their CQ-sensitive counterparts. This may explain in part why antagonism was intensified in CQ-resistant parasites. However, equimolar treatment with 10 µM CQ resulted in significantly lower Ac-H-FluNox signal in Dd2 PfCRT$^{Dd2}$ versus Dd2 PfCRT$^{3D7}$ parasites (*Figure 3f*). These data could suggest that while the majority of CQ is sequestered in the digestive vacuole of Dd2 PfCRT$^{3D7}$ parasites, PfCRT$^{Dd2}$ is able to transport more CQ out of the digestive vacuole, thus allowing more CQ to bind cytosolic heme in parasites harboring a PfCRT$^{Dd2}$. This duality of PfCRT$^{Dd2}$, simultaneously enhancing parasite survival to CQ while promoting cytoplasmic heme 'inactivation', may explain the unprecedented one-sided superantagonistic isobole that we observed.

Fortunately, we did not observe superantagonism with DHA-PPQ in PPQ-resistant parasites. With a modest threefold shift in PPQ IC50 between Dd2 and Dd2 PfCRT$^{M343L}$ parasites (25 nM versus 65 nM), it is perhaps unsurprising that no significant difference in DHA-PPQ isobolograms was observed between these two parasites. This is in contrast to the >1000-fold CQ IC50 shift that we and others (*Gligorijevic et al., 2008*) have observed between CQ-sensitive and CQ-resistant parasites using pulsing assays. Our studies with Ac-H-FluNox indicate that 150 nM PPQ is sufficient to render approximately 50% of available heme inert (*Figure 3e*). While these data suggest that PPQ blocks some DHA activation, it is likely that 150 nM PPQ is not sufficient to block all DHA activation. Since even PPQ-resistant parasites are at least partially susceptible to mid-nanomolar PPQ concentrations, the protective effect that PPQ has in neutralizing DHA is partially negated by PPQ-mediated killing. In contrast, CQ-resistant parasites are relatively unaffected by pulses with low micromolar CQ concentrations. Accordingly, CQ concentrations that are high enough to nearly block all DHA activation are insufficient to kill CQ-resistant parasites.

Relative to PPQ and MFQ, CQ appears to have the greatest ability to impair heme iron reactivity. We were surprised to observe that PPQ inhibited H-FluNox fluorescence only half as well as CQ in solution. This suggests that there may be important differences in how these 4-aminoquinolines bind heme. In Dd2 parasites, CQ also inhibited Ac-H-FluNox fluorescence to a greater extent than PPQ at 10 µM concentrations. Given that the ability of CQ to 'inactivate' heme in parasites is dependent on PfCRT genotype, it is possible that these results may differ in a PPQ-resistant parasite. In solution, equimolar concentrations of CQ and MFQ inhibited H-FluNox fluorescence to a similar extent, while MFQ had a minimal effect on Ac-H-FluNox fluorescence at even super-pharmacological concentrations. This was somewhat surprising given that MFQ is believed to act in the parasite cytoplasm (*Sidhu et al., 2006*; *Rohrbach et al., 2006*). It's possible that differences in how MFQ binds to heme (*Dorn et al., 1998*), affinity of MFQ for other biomolecules (*Wong et al., 2017*), and/or modification of MFQ within the parasite could contribute to this discrepancy.

In the early 2000s, the rise of CQ resistance prompted a search for new combination therapies (*Packard, 2014*). Several clinical trials were conducted in West Africa to compare the efficacy of CQ versus CQ-AS. On day 28, two studies reported similar failure rates for CQ and CQ-AS (*Sutherland et al., 2003*; *Kofoed et al., 2003*). In a third study, cure rates were significantly higher for CQ-AS compared to CQ alone (*Sirima et al., 2003*). However, all three studies recommended against use of CQ-AS, due to poor treatment efficacy. Notably, AS-SP and AS-LM were reported to show >97% efficacy on West African populations during the same time period (*von Seidlein et al., 2001*; *Mutabingwa et al., 2005*). In contrast to superantagonism, in vitro conventional antagonism does not appear to correlate well with treatment failure. ACTs, including DHA-PPQ and AS-ADQ, are highly effective, provided that parasites remain sensitive to one or both ACT component(s). It has yet to be evaluated if antagonistic combinations have silent consequences that might discourage their clinical implementation. For example, since PPQ and ADQ appear to block artemisinin activation, it is possible that these combinations are acting more like pseudo-monotherapies than true combination therapies. This could have a number of downstream consequences, such as impacting the rate of drug resistance evolution. However, future work is needed to address these concerns.

With the absence of new antimalarials ready for clinical implementation, novel strategies with existing drugs have been proposed to overcome current problems with ACT treatment failures. One

such strategy, TACTs, has gained the most traction. For traditional ACTs, the short 1 hr half-life of artemisinins means that long-lived partner drugs are essentially acting as monotherapies for most of the initial 3-day treatment regimen and a substantial period thereafter. In the case of artemisinin-resistant parasites, partner drugs must clear a significantly higher parasite biomass, which increases the likelihood for resistance to emerge. In theory, TACTs are designed to overcome this flaw, whereby two partner drugs with similar half-lives are used, ensuring that parasites are always exposed to at least two antimalarials (*Nguyen et al., 2023*; *van der Pluijm et al., 2021*). In clinical trials, DHA-PPQ-MFQ showed improved efficacy over DHA-PPQ in regions of high PPQ resistance (*van der Pluijm et al., 2020*). However, some have speculated that this was solely due to the application of MFQ to MFQ-sensitive parasites and that the TACT would be no more effective than DHA-MFQ (*Wang et al., 2021*). PPQ and MFQ were initially promoted as ideal partner drugs for TACTs as it was thought that either due to collateral sensitivity or fitness costs, dual PPQ and MFQ resistance was unlikely to occur (*Amato et al., 2017*; *Witkowski et al., 2017*). Use of edited isogenic lines suggests that PPQ resistance-conferring mutations in PfCRT have no effect on MFQ sensitivity, while increased MDR1 copy number, which underlies MFQ resistance, has no effect on PPQ sensitivity (*Dhingra et al., 2017*). Further, more recent genetic surveillance studies of Cambodian parasites cast doubt on the hypothesis that a DHA-PPQ-MFQ-resistant parasite would be unfit to spread clinically (*Rossi et al., 2017*). Previous reports using standard 48 hr or 72 hr assays indicated that PPQ and MFQ are modestly antagonistic (*Davis et al., 2006*; *Ansbro et al., 2020*). Interestingly, we observed a much more pronounced super-antagonistic phenotype using a pulsing format. It is important to note that unlike artemisinins, PPQ and MFQ have half-lives of several weeks (*Palmer et al., 1993*; *Rijken et al., 2011*), raising the question of which assay format would be best suited to evaluate interactions of long-lived partner drugs. Nevertheless, given the interplay of existing PPQ resistance and PPQ-MFQ antagonism, MFQ and PPQ should, perhaps, be further scrutinized as potential partners for TACTs.

The synthetic ozonide, OZ439, was previously poised to replace artemisinins in combination therapies. While OZ439 is believed to have a mechanism of action similar to that of artemisinins, it has a notably improved 46–62 hr half-life (*Phyo et al., 2016b*) and was shown to be effective against artemisinin-resistant parasites (*Walz et al., 2019*; *Baumgärtner et al., 2017*; *Straimer et al., 2017*). While OZ439 showed good efficacy alone (*Phyo et al., 2016b*), high treatment failure as part of single-dose combination therapies ultimately resulted in a loss of interest in this antimalarial (*Macintyre et al., 2017*; *Adoke et al., 2021*; *Gansane et al., 2023*). However, our data suggest that unsuitable choices in partner drugs may be one of several factors, including poor drug exposure (*Macintyre et al., 2017*; *Adoke et al., 2021*; *Gansane et al., 2023*) and preexisting PPQ resistance (*Hamilton et al., 2019*; *Phuc et al., 2017*), that contributed to the poor therapeutic efficacy of these combinations. As assessed by Ac-H-FluNox, both PPQ and FQ appear to render heme inert, thus blocking OZ439 activation. Biochemical studies have predicted that due to its less basic behavior, FQ might have a lower propensity to accumulate in the parasite digestive vacuole compared to CQ (*Biot et al., 2005*). Greater accumulation of FQ in the parasite cytoplasm compared to other quinolines might explain the ability of FQ to greatly diminish Ac-H-FluNox signal even at nanomolar concentrations. Clinically, peak levels of FQ have been reported to reach approximately 0.5 µM (*Supan et al., 2017*), which could significantly impact OZ439 activation. However, unlike CQ, mid-nanomolar FQ concentrations are lethal for parasites, which may explain why we did not observe a superantagonistic phenotype for FQ-OZ439.

The antimalarials that we have are a precious resource, and it is only a matter of time before they fail. Our work clearly demonstrates the importance of detailed understanding of drug-drug interactions at the parasite level. While we believe the pulsing assay used in this study better mimics artemisinin pharmacokinetics compared to standard 72 hr assays, it is not suitable for high-throughput format. Recently, a high-throughput assay was developed to evaluate drug-drug and drug-drug-drug interactions in *P. falciparum* (*Ansbro et al., 2020*). Using this assay, the authors were able to evaluate interactions of 16 potential TACT combinations against 13 clinical isolates (*Ansbro et al., 2020*). Similar to our findings, the authors reported that interactions can be highly dependent on parasite genetics. This high-throughput assay will undoubtedly be an important first step in evaluating interactions of new combinations. Nevertheless, since this assay relies on a 72 hr format, additive interactions should be interpreted with caution, as this format can mask drug-drug interactions (*Stokes et al., 2019*). Accordingly, pulsing assays like the one used in this study will be

an important step for investigating drug-drug interactions, prior to in vivo studies (*Demarta-Gatsi et al., 2023*).

## Materials and methods

### Reagents

All reagents were purchased from Sigma-Aldrich unless otherwise indicated.

### Parasite culture

Dd2 PfCRT$^{Dd2}$, Dd2PfCRT$^{3D7}$, Dd2 PfCRT$^{M343L}$, and Dd2 K13$^{R539T}$ were kindly provided by Dr. David Fidock (Columbia University) (*Straimer et al., 2015*; *Ross et al., 2018*). MRA-1284 was obtained through BEI Resources, NIAID, NIH: *P. falciparum*, Strain IPC_6261, MRA-1284, contributed by Dr. Didier Ménard. Parasites, which were confirmed *Mycoplasma* negative by sequencing and resistance testing, were propagated in human red blood cells obtained from St. Louis Children's Hospital blood bank. Cultures were grown at 5% hematocrit in media containing RPMI1640 (Gibco) supplemented with 0.25% (wt/vol) Albumax (Gibco), 15 mg/L hypoxanthine, 110 mg/L sodium pyruvate, 1.19 g/L HEPES, 2.52 g/L sodium bicarbonate, 2 g/L glucose, and 10 mg/L gentamicin and were maintained in air-tight chambers filled with 5% $O_2$/5% $CO_2$/90% $N_2$ at 37°C.

### Trophozoite-stage dose-response assays and isobolograms

Trophozoite-stage assays were performed as previously described with minor modifications (*Rosenthal and Ng, 2021*). Briefly, parasites were synchronized to the trophozoite stage using two consecutive treatments with 5% sorbitol 10 hr apart, followed by culturing for an additional 14 hr. Trophozoite stages were seeded at 0.2% parasitemia and 1% hematocrit in round-bottom 96-well plates. Following a 4 hr incubation with antimalarials, infected red blood cells were washed a minimum of five times for plates containing OZ439 and four times for all other antimalarials. Plates were centrifuged, 190 µL of media was carefully removed, and 190 µL of fresh media was added. Following the last wash, cultures were transferred to a new flat-bottom 96-well plate. After 72 hr, parasitemia was assessed by flow cytometry. Cultures were resuspended, and 10 µL was transferred to a new 96-well plate containing 50 µL per well of 1× PBS with 1× SYBR Green I and 100 nM MitoTracker Deep Red. Following incubation at 37°C for 1 hr, plates were read on a BD FACSCanto with a BD HTS automated plate reader and analyzed with BD FACSDiva Software version 8. Approximately 10,000 events were analyzed per well. GraphPad version 10 was used to calculate IC50 values by nonlinear regression.

For dose-response assays with fixed concentrations of partner drugs, a 96-well plate was prepared containing twofold serial dilutions of DHA media at 100 µL per well. A bulk intermediate culture of parasites was prepared at 2% hematocrit. Partner drugs were added to this intermediate culture at 2× concentrations. After mixing, 100 µL of the intermediate culture was added to the 96-well plate containing DHA serial dilutions, for a final hematocrit of 1% and a 1× fixed partner drug concentration.

For isobolograms, the following fixed ratios were prepared as starting concentrations: 1:0, 4:1, 2:1, 1:1, 1:2, 1:4, 0:1. The starting concentration of each drug alone (1:0 and 0:1 ratios) represents an approximate 16× IC50 concentration. Trophozoite stages were exposed to twofold dilutions of these ratios. Fractional IC50 (FIC50) values were calculated as follows:

$$FIC50 = \frac{IC50 \text{ of drugs in combination}}{IC50 \text{ of individual drug}}$$

FIC50 values were plotted on an XY graph. Mean ΣFIC50 values were calculated by averaging the sums of FIC50 values from 4:1, 2:1, 1:1, 1:2, and 1:4 ratios. Trophozoite-stage IC50 values for *Figure 1—figure supplement 1* and *Figure 4—figure supplement 2* were obtained from 1:0 or 0:1 ratios. A one-way ANOVA with a Tukey's or Dunnett's multiple comparison test was used to assess statistical significance for IC50 values.

### RSAs and early ring-stage dose-response assays

0–3 hr post invasion (hpi) rings were obtained as previously described (*Hasan et al., 2023*). Briefly, schizonts were enriched using MACS LD magnet columns (Miltenyi Biotec). Schizont-enriched cultures were diluted to 2% hematocrit in 3–4 mL of media and incubated for 3 hr with shaking. A 5% sorbitol

treatment was then used to remove remaining mature parasites. Early ring stages were seeded at 1% parasitemia and 1% hematocrit in round-bottom 96-well plates. Twofold serial dilutions were prepared starting at 1400 nM DHA, so that both IC50 values and RSA values could be obtained. After incubation for 3 hr, antimalarials were washed out, and parasitemia was assessed 72 hr later by flow cytometry as described in the previous section. GraphPad Prism version 10 was used to calculate IC50 values by nonlinear regression. RSA survival was obtained by dividing the parasitemia of cultures treated with 700 nM DHA by the untreated control. A one-way ANOVA with a Dunnett's multiple comparison test was used to determine statistical significance for IC50 values and RSA survival values.

## Detection of free heme in vitro

H-FluNox was kindly provided by Dr. Tasuku Hirayama (Gifu Pharmaceutical University, Lab of Pharmaceutical & Medicinal Chemistry). Reactions were carried out in a buffer containing 50 mM HEPES (pH 7.4) and 100 µM reduced glutathione. Glutathione was used to convert hemin into heme. Hemin was made up in reaction buffer, and twofold serial dilutions were performed to obtain a range of heme concentrations starting at 500 nM. H-FluNox was used at a final concentration of 500 nM. Reactions were performed in black-walled clear-bottom 96-well plates. Following incubation with H-FluNox and indicated antimalarials at room temperature for 30 min, fluorescence was read at 485/535 nm ex/em using an EnVision Multimode Plate Reader.

## Detection of free heme in live parasites

Ac-H-FluNox was kindly provided by Dr. Tasuku Hirayama (Gifu Pharmaceutical University, Lab of Pharmaceutical & Medicinal Chemistry). Parasites were synchronized to the trophozoite stage as described above. Cultures were then treated with the indicated compound for the indicated time. Media was removed, and parasite cultures were resuspended and incubated in 1× PBS containing 10 µM Ac-H-FluNox for 30 min at 37°C. Ac-H-FluNox is the cell-permeable analog of H-FluNox and is converted to H-FluNox in cells (*Kawai et al., 2022*). Following incubation, cultures were washed with 1× PBS and then imaged using a 63× objective on a Zeiss Imager M2 Plus Widefield fluorescence microscope with AxioVision 4.8 software. Except for *Figure 3—figure supplement 1b*, a fixed exposure of 350 ms was used to capture H-FluNox fluorescence at 470/509 nm ex/em. Exposure for *Figure 3—figure supplement 1b* was adjusted for each condition so that DV staining could be optimally visualized. For quantification in *Figures 3e, f and 4i*, *Figure 3—figure supplement 2*, at least 10 fields were captured per treatment condition and then mean fluorescence intensity of parasite minus background was quantified with ImageJ. Statistical significance of ≥70 parasites from at least three independent drug treatments was assessed using a one-way ANOVA.

## Western blot

Parasite drug treatment, protein harvest, sample preparation, SDS-PAGE, and wet transfer were performed as previously described (*Rosenthal et al., 2024b*). Blots were blocked with 3% BSA in TBS-T and incubated overnight with primary antibodies used at 1:1000 in 1× TBS-T. Anti K48-linked ubiquitin was obtained from Cell Signaling Technologies (catalog number: D9D5, rabbit). Anti-PMV was obtained from *Banerjee et al., 2002*. Blots were incubated for 1 hr at room temperature with IRDye-conjugated goat secondary antibodies (LI-COR) at 1:10,000 dilutions. Blots were imaged using a Bio-Rad imager with Image Lab Touch Software and then quantified with ImageJ. Significant differences between control and antimalarial-treated parasites were determined using a one-way ANOVA with a Dunnett's multiple comparison test.

## Acknowledgements

Funding for this work was provided by T32AI007172 and F32AI188627 (to MRR). We thank David Fidock (Columbia University) and Miles Siegel (LGENIA) for helpful discussion.

## Additional information

### Funding

| Funder | Grant reference number | Author |
|---|---|---|
| National Institute of Allergy and Infectious Diseases | F32 AI188627 | Melissa Rosenthal |
| National Institute of Allergy and Infectious Diseases | T32AI007172 | Melissa Rosenthal |

The funders had no role in study design, data collection and interpretation, or the decision to submit the work for publication.

### Author contributions

Melissa Rosenthal, Conceptualization, Data curation, Formal analysis, Funding acquisition, Validation, Investigation, Visualization, Methodology, Writing – original draft, Writing – review and editing; Daniel E Goldberg, Conceptualization, Resources, Data curation, Formal analysis, Supervision, Funding acquisition, Investigation, Methodology, Writing – original draft, Project administration, Writing – review and editing

### Author ORCIDs

Melissa Rosenthal (iD) https://orcid.org/0000-0002-4490-6010
Daniel E Goldberg (iD) https://orcid.org/0000-0003-3529-8399

Reviewer #1 (Public review): https://doi.org/10.7554/eLife.108976.3.sa1
Reviewer #2 (Public review): https://doi.org/10.7554/eLife.108976.3.sa2
Reviewer #3 (Public review): https://doi.org/10.7554/eLife.108976.3.sa3
Author response https://doi.org/10.7554/eLife.108976.3.sa4

## Additional files

### Supplementary files
MDAR checklist

### Data availability
All data generated or analyzed during this study are included in the manuscript and supporting files; source data files have been provided.

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
