## [Editor Report · eLife Assessment]

This is an **important** study with direct implications for the rational selection of antimalarial drug combinations. The authors present data demonstrating antagonism between 4-aminoquinoline antimalarials and peroxide drugs under physiologically relevant conditions, including robust effects at the trophozoite stage and for chloroquine at the ring stage. While the conclusions are based on in vitro assays and further work will be needed to fully resolve the underlying mechanism, the findings are **convincing** and provide a strong rationale for evaluating drug combinations in relevant preclinical models prior to clinical testing.

---

## [Referee Report · Reviewer #1 (Public review)]

Summary:

This study set out to investigate potential pharmacological drug-drug interactions between the two most common antimalarial classes, the artemisinins and quinolines. There is strong rationale for this aim, because drugs from these classes are already widely-used in Artemisinin Combination Therapies (ACTs) in the clinic, and drug combinations are an important consideration in the development of new medicines. Furthermore, whilst there is ample literature proposing many diverse mechanisms of action and resistance for the artemisinins and quinolines, it is generally accepted that the mechanisms for both classes involve heme metabolism in the parasite, and that artemisinin activity is dependent on activation by reduced heme. The study was designed to measure drug-drug interactions associated with a short pulse exposure (4 h) that is reminiscent of the short duration of artemisinin exposure obtained after in vivo dosing. Clear antagonism was observed between dihydroartemisinin (DHA) and chloroquine, which became even more extensive in chloroquine-resistant parasites. Antagonism was also observed in this assay for the more clinically-relevant ACT partner drugs piperaquine and amodiaquine, but not for other ACT partners mefloquine and lumefantrine, which don't share the 4-aminoquinoline structure or mode of action. Interestingly, chloroquine induced an artemisinin resistance phenotype in the standard in vitro Ring-stage Survival Assay, whereas this effect was not as extensive for piperaquine.

The authors also utilised a heme-reactive probe to demonstrate that the 4-aminoquinolines can inhibit heme-mediated activation of the probe within parasites, which suggests that the mechanism of antagonism involves the inactivation of heme, rendering it unable to activate the artemisinins. Measurement of protein ubiquitination showed reduced DHA-induced protein damage in the presence of chloroquine, which is also consistent with decreased heme-mediated activation, and/or with decreased DHA activity more generally.

Overall, the study clearly demonstrates a mechanistic antagonism between DHA and 4-aminoquinoline antimalarials in vitro. It is interesting that this combination is successfully used to treat millions of malaria cases every year, which may raise questions about the clinical relevance of this finding. However, the conclusions in this paper are supported by multiple lines of evidence and the data is clearly and transparently presented, leaving no doubt that DHA activity is compromised by the presence of chloroquine in vitro. It is perhaps fortunate the that the clinical dosing regimens of 4-aminoquinoline-based ACTs have been sufficient to maintain clinical efficacy despite the non-optimal combination. Nevertheless, optimisation of antimalarial combinations and dosing regimens is becoming more important in the current era of increasing resistance to artemisinins and 4-aminoquinolines. Therefore, these findings should be considered when proposing new treatment regimens (including Triple-ACTs) and the assays described in this study should be performed on new drug combinations that are proposed for new or existing antimalarial medicines.

Strengths:

This manuscript is clearly written and the data presented is clear and complete. The key conclusions are supported by multiple lines of evidence, and most findings are replicated with multiple drugs within a class, and across multiple parasite strains, thus providing more confidence in the generalisability of these findings across the 4-aminoquinoline and peroxide drug classes.

A key strength of this study was the focus on short pulse exposures to DHA (4 h in trophs and 3 h in rings), which is relevant to the in vivo exposure of artemisinins. Artemisinin resistance has had a significant impact on treatment outcomes in South-East Asia, and is now emerging in Africa, but is not detected using a 'standard' 48 or 72 h in vitro growth inhibition assay. It is only in the RSA (a short pulse of 3-6 h treatment of early ring stage parasites) that the resistance phenotype can be detected in vitro. Therefore, assays based on this short pulse exposure provide the most relevant approach to determine whether drug-drug interactions are likely to have a clinically-relevant impact on DHA activity. These assays clearly showed antagonism between DHA and 4-aminoquinolines (chloroquine, piperaquine, amodiaquine and ferroquine) in trophozoite stages. Interestingly, whilst chloroquine clearly induced an artemisinin-resistant phenotype in the RSA, piperaquine only had a minor impact on the early ring stage activity of DHA, which may be fortunate considering that piperaquine is a currently recommended DHA partner drug in ACTs, whereas chloroquine is not.

The evaluation of additional drug combinations at the end of this paper is a valuable addition, which increases the potential impact of this work. The finding of antagonism between piperaquine and OZ439 in trophozoites is consistent with the general interactions observed between peroxides and 4-aminoquinolines, and it may be interesting to see whether piperaquine impacts the ring-stage activity of OZ439.

The evaluation of reactive heme in parasites using a fluorescent sensor, combined with the measurement of K48-linked ubiquitin, further support the findings of this study, providing independent read-outs for the chloroquine-induced antagonism.

The in-depth discussion of the interpretation and implications of the results are an additional strength of this manuscript. Whilst the discussion section is rather lengthy, there are important caveats to the interpretation of some of these results, and clear relevance to the future management of malaria that require these detailed explanations.

Overall, this is a high quality manuscript describing an important study that has implications for the selection of antimalarial combinations for new and existing malaria medicines.

Weaknesses:

This study is an in vitro study of parasite cultures, and therefore caution should be taken when applying these findings to decisions about clinical combinations. The drug concentrations and exposure durations in these assays are intended to represent clinically relevant exposures, although it is recognised that the in vitro system is somewhat simplified and there may be additional factors that influence in vivo activity. This limitation is reasonably well acknowledged in the manuscript.

It is also important to recognise that the majority of the key findings regarding antagonism are based on trophozoite-stage parasites, and one must show caution when generalising these findings to other stages or scenarios. For example, piperaquine showed clear antagonism in trophozoite stages, but minimal impact in ring stages under these assay conditions.

A key limitation is the interpretation of the mechanistic studies that implicate heme-mediated artemisinin activation as the mechanism underpinning antagonism by chloroquine. This study did not directly measure the activation of artemisinins. The data obtained from the activation of the fluorescent probe are generally supportive of chloroquine suppressing the heme-mediated activation of artemisinins, and I think this is the most likely explanation, but there are significant caveats to consider. Primarily, the inconsistency between the fluorescence profile in the chemical reactions and the cell-based assay raise questions about the accuracy of this readout. In the chemical reaction, mefloquine and chloroquine showed identical inhibition of fluorescence, whereas piperaquine had minimal impact. On the contrary, in the cell, chloroquine and piperaquine had similar impacts on fluorescence, but mefloquine had minimal impact. This inconsistency indicates that the cellular fluorescence based on this sensor does not give a simple direct readout of the reactivity of ferrous heme, and therefore, these results should be interpreted with caution. Indeed, the correlation between fluorescence and antagonism for the tested drugs is a correlation, not causation. There could be several reasons for the disconnect between the chemical and biological results, either via additional mechanisms that quench fluorescence, or the presence of biomolecules that alter the oxidation state or coordination chemistry of heme or other potential catalysts of this sensor. It is possible that another factor that influences the H-FluNox fluorescence in cells also influences the DHA activity in cells, leading to the correlation with activity. It should be noted that H-FluNox is not a chemical analogue of artemisinins. It's activation relies on Fenton-like chemistry, but with a N-O rather that O-O bond, and it possesses very different steric and electronic substituents around the reactive centre, which are known to alter reactivity to different iron sources. Despite these limitations, the authors have provided reasonable justification for the use of this probe to directly visualise heme reactivity in cells, and the results are still informative.

Another interesting finding that was not elaborated by the authors is the impact of chloroquine in the DHA dose-response curves from the ring stage assays. Detection of artemisinin resistance in the RSA generally focuses on the % survival at high DHA concentrations (700 nM) as there is minimal shift in the IC50 (see Fig 2), however, chloroquine clearly induces a shift in the IC50 (~5-fold), where the whole curve is shifted to the right, whereas the increase in % survival is relatively small. This different profile suggests that the mechanism of chloroquine-induced antagonism may be different to the mechanism of artemisinin resistance. Current evidence regarding the mechanism of artemisinin resistance generally points towards decreased heme-mediated drug activation due to a decrease in hemoglobin uptake, which should be analogous to the decrease in heme-mediated drug activation caused by chloroquine. However, these different dose response curves suggest different mechanisms are primarily responsible. Additional mechanisms have been proposed for artemisinin resistance, involving redox or heat stress responses, proteostatic responses, mitochondrial function, dormancy and PI3K signalling among others. Whilst the H-FluNox probe generally supports the idea that chloroquine suppresses heme-mediated DHA activation, it remains plausible that chloroquine could induce these, or other, cellular responses that suppress DHA activity.

Impact:

This study has important implications for the selection of drugs to form combinations for the treatment of malaria. The overall findings of antagonism between peroxide antimalarials and 4-aminoquinolines in the trophozoite stage are robust, and the this carries across to the ring stage for chloroquine.

The manuscript also provides a plausible mechanism to explain the antagonism, although future work will be required to further explore the details of this mechanism and to rule out alternative factors that may contribute.

Overall, this is an important contribution to the field and provides a clear justification for the evaluation of potential drug combinations in relevant in vitro assays before clinical testing.

---

## [Referee Report · Reviewer #2 (Public review)]

Summary:

This manuscript by Rosenthal and Goldberg investigates interactions between artemisinins and its quinoline partner drugs currently used for treating uncomplicated *Plasmodium falciparum* malaria. The authors show that chloroquine (CQ), piperaquine, and amodiaquine antagonize dihydroartemisinin (DHA) activity, and in CQ-resistant parasites, the interaction is described as "superantagonism," linked to the pfcrt genotype. Mechanistically, application of the heme-reactive probe H-FluNox indicates that quinolines render cytosolic heme chemically inert, thereby reducing peroxide activation. The work is further extended to triple ACTs and ozonide-quinoline combinations, with implications for artemisinin-based combination therapy (ACT) design, including triple ACTs.

Strengths:

The manuscript is clearly written, methodologically careful, and addresses a clinically relevant question. The pulsing assay format more accurately models in vivo artemisinin exposure than conventional 72-hour assays, and the use of H-FluNox and Ac-H-FluNox probes provides mechanistic depth by distinguishing chemically active versus inert heme. These elements represent important refinements beyond prior studies, adding nuance to our understanding of artemisinin-quinoline interactions.

Weaknesses:

Several points warrant consideration. The novelty of the work is somewhat incremental, as antagonism between artemisinins and quinolines is well established. Multiple prior studies using standard fixed-ratio isobologram assays have shown that DHA exhibits indifferent or antagonistic interactions with chloroquine, piperaquine, and amodiaquine (e.g., Davis et al., 2006; Fivelman et al., 2007; Muangnoicharoen et al., 2009), with recent work highlighting the role of parasite genetic background, including pfcrt and pfmdr1, in modulating these interactions (Eastman et al., 2016). High-throughput drug screens likewise identify quinoline-artemisinin combinations as mostly antagonistic. The present manuscript adds refinement by applying pulsed-exposure assays and heme probes rather than establishing antagonism de novo.

The dataset focuses on several parasite lines assayed in vitro, so claims about broad clinical implications should be tempered, and the discussion could more clearly address how in vitro antagonism may or may not translate to clinical outcomes. The conclusion that artemisinins are predominantly activated in the cytoplasm is intriguing but relies heavily on Ac-H-FluNox data, which may have limitations in accessing the digestive vacuole and should be acknowledged explicitly. The term "superantagonism" is striking but may appear rhetorical; clarifying its reproducibility across replicates and providing a mechanistic definition would strengthen the framing. Finally, some discussion points, such as questioning the clinical utility of DHA-PPQ, should be moderated to better align conclusions with the presented data while acknowledging the complexity of in vivo pharmacology and clinical outcomes.

Despite these mild reservations, the data are interesting and of high quality and provide important new information for the field.

Editor's Review of the Revision: The authors have provided a well-reasoned rebuttal to the comments of the three reviewers. Most of the changes were incorporated in their revised Discussion. Their data with the active heme probe H-FluNox are novel and the authors reveal interesting interactions between peroxide and 4-aminoquinoline-based antimalarials that open new avenues of research especially when considering antimalarial combinations that combine these chemical scaffolds. This study will be of broad interest to investigators studying and developing antimalarial drugs and combinations and the impact of *Plasmodium falciparum* resistance mechanisms. A minor recommendation would be that the authors state H-FluNox when referring to their small molecule probe in the abstract, so that it is captured in PubMed searches.

---

## [Referee Report · Reviewer #3 (Public review)]

Summary:

The authors present an in vitro evaluation of drug-drug interactions between artemisinins and quinoline antimalarials, as an important aspect for screening the current artemisinin-based combination therapies for *Plasmodium falciparum*. Using a revised pulsing assay, they report antagonism between dihydroartemisinin (DHA) and several quinolines, including chloroquine, piperaquine (PPQ), and amodiaquine. This antagonism is increased in CQ-resistant strains in isobologram analyses. Moreover, CQ co-treatment was found to induce artemisinin resistance even in parasites lacking K13 mutations during the ring-stage survival assay. This implies that drug-drug interactions, not just genetic mutations, can influence resistance phenotypes. By using a chemical probe for reactive heme, the authors demonstrate that quinolines inhibit artemisinin activation by rendering cytosolic heme chemically inert, thereby impairing the cytotoxic effects of DHA. The study also observed negative interactions in triple-drug regimens (e.g., DHA-PPQ-Mefloquine) and in combinations involving OZ439, a next-generation peroxide antimalarial. Taken together, these findings raise significant concerns regarding the compatibility of artemisinin and quinoline combinations, which may promote resistance or reduce efficacy.

With the additive profile as the comparison and a lack of synergistic effect in any of the comparisons, it is hard to contextualize the observed antagonism. Including a known synergistic pair (e.g., artemisinin + lumefantrine) would have provided a useful benchmark to assess the relative impact of the drug interactions described.

Strengths:

This study demonstrates the following strengths:

• The use of a pulsed in vitro assay that is more physiologically relevant over the traditional 48h or 72h assays

• Small molecule probes, H-FluNox, and Ac-H-FluNox to detect reactive cytosolic heme, demonstrating that quinolines render heme inert and thereby block DHA activation.

• Evaluates not only traditional combinations but also triple-drug combinations and next-generation artemisinins like OZ439. This broad scope increases the study's relevance to current treatment strategies and future drug development.

• By using the K13 wild-type parasites, the study suggests that resistance phenotypes can emerge from drug-drug interactions alone, without requiring genetic resistance markers.

Weaknesses:

• The study would benefit from a future characterization of the molecular basis for the observed heme inactivation by quinolines to support this hypothesis - while the probe experiments are valuable, they do not fully elucidate how quinolines specifically alter heme chemistry at the molecular level.

• Suggestion of alternative combinations that show synergy could have improved the significance of the work. The invitro study did not include pharmacokinetic/pharmacodynamic modeling, hence it leaves questions about how the observed antagonism would manifest under real-world dosing conditions, necessitating furture work based on these findings.

---

## [Author Response]

The following is the authors’ response to the original reviews.

**eLife Assessment**

We appreciate the positive assessment. We recognize that since all of the work in this manuscript was done in vitro, there are reasonable concerns about the translatability of these data to clinical settings. These results should not directly inform malaria policy, but we hope that these data bring new considerations to the approach for choosing strategic antimalarial combinations. We have modified the manuscript to clarify this distinction.

**Public Reviews**

**Reviewer #1 (Public Review):**

We thank the reviewer for their thoughtful summary of this manuscript. It is important to note that DHA-PPQ did show antagonism in RSAs. In this modified RSA, 200 nM PPQ alone inhibited growth of PPQ-sensitive parasites approximately 20%. If DHA and PPQ were additive, then we would expect that addition of 200 nM PPQ would shift the DHA dose response curve to the left and result in a lower DHA IC50. Please refer to Figure 4a and b as examples of additive relationships in dose-response assays. We observed no significant shift in IC50 values between DHA alone and DHA + PPQ. This suggests antagonism, albeit not to the extent seen with CQ. We have modified the manuscript to emphasize this point. As the reviewer pointed out, it is fortunate that despite being antagonistic, clinically used artemisinin-4-aminoquinoline combinations are effective, provided that parasites are sensitive to the 4-aminoquinoline. It is possible that superantagonism is required to observe a noticeable effect on treatment efficacy (Sutherland et al. 2003 and Kofoed et al. 2003), but that classical antagonism may still have silent consequences. For example, if PPQ blocks some DHA activation, this might result in DHA-PPQ acting more like a pseudo-monotherapy. However, as the reviewer pointed out, while our data suggest that DHA-PPQ and AS-ADQ are “non-optimal” combinations, the clinical consequences of these interactions are unclear. We have modified the manuscript to emphasize the later point.

While the Ac-H-FluNox and ubiquitin data point to a likely mechanism for DHA-quinoline antagonism, we agree that there are other possible mechanisms to explain this interaction. We have addressed this limitation in the discussion section. Though we tried to measure DHA activation in parasites directly, these attempts were unsuccessful. We acknowledge that the chemistry of DHA and Ac-H-FluNox activation is not identical and that caution should be taken when interpreting these data. Nevertheless, we believe that Ac-H-FluNox is the best currently available tool to measure “active heme” in live parasites and is the best available proxy to assess DHA activation in live parasites. These points are now addressed in the discussion section. Both in vitro and in parasite studies point to a roll for CQ in modulating heme, though an exact mechanism will require further examination. Similar to the reviewer, we were perplexed by the differences observed between in vitro and in parasite assays with PPQ and MFQ. We proposed possible hypotheses to explain these discrepancies in the discussion section. Interestingly, our data corelate well with hemozoin inhibition assays in which all three antimalarials inhibit hemozoin formation in solution, but only CQ and PPQ inhibit hemozoin formation in parasites. In both assays, in-parasite experiments are likely to be more informative for mechanistic assessment.

It remains unclear why K13 genotype influences RSA values, but not early ring DHA IC50 values. In K13^WT^ parasites, both RSA values and DHA IC50 values were increased 3-5 fold upon addition of CQ. This suggests that CQ-mediated resistance is more robust than that conferred by K13 genotype. However, this does not necessarily suggest a different resistance mechanism. We acknowledge that in addition to modulating heme, it is possible that CQ may enhance DHA survival by promoting parasite stress responses. Future studies will be needed to test this alternative hypothesis. This limitation has been acknowledged in the manuscript. We have also addressed the reviewer’s point that other factors, including poor pharmacokinetic exposure, contributed to OZ439-PPQ treatment failure.

**Reviewer #2 (Public Review):**

We appreciate the positive feedback. We agree that there have been previous studies, many of which we cited, assessing interactions of these antimalarials. We also acknowledge that previous work, including our own, has shown that parasite genetics can alter drug-drug interactions. We have included the author’s recommended citations to the list of references that we cited. Importantly, our work was unique not only for utilizing a pulsing format, but also for revealing a superantagonistic phenotype, assessing interactions in an RSA format, and investigating a mechanism to explain these interactions. We agree with the reviewer that implications from this in vitro work should be cautious, but hope that this work contributes another dimension to critical thinking about drug-drug interactions for future combination therapies. We have modified the manuscript to temper any unintended recommendations or implications.

The reviewer notes that we conclude “artemisinins are predominantly activated in the cytoplasm”. We recognize that the site of artemisinin activation is contentious. We were very clear to state that our data combined with others suggest that artemisinins can be activated in the parasite cytoplasm. We did not state that this is the primary site of activation. We were clear to point out that technical limitations may prevent Ac-H-FluNox signal in the digestive vacuole, but determined that low pH alone could not explain the absence of a digestive vacuole signal.

With regard to the “reproducibility” and “mechanistic definition” of superantagonism, we observed what we defined as a one-sided superantagonistic relationship for three different parasites (Dd2, Dd2 PfCRT^Dd2^, and Dd2 K13^R539T^) for a total of nine independent replicates. In the text, we define that these isoboles are unique in that they had mean ΣFIC50 values > 2.4 and peak ΣFIC50 values >4 with points extending upward instead of curving back to the axis. As further evidence of the reproducibility of this relationship, we show that CQ has a significant rescuing effect on parasite survival to DHA as assessed by RSAs and IC50 values in early rings.

**Reviewer #3 (Public Review):**

We thank the reviewer for their positive feedback. We acknowledge that no combinations tested in this manuscript were synergistic. However, two combinations, DHA-MFQ and DHA-LM, were additive, which provides context for contextualizing antagonistic relationships. We have previously reported synergistic and additive isobolograms for peroxide-proteasome inhibitor combinations using this same pulsing format (Rosenthal and Ng 2021). These published results are now cited in the manuscript.

We believe that these findings are specific to 4-aminoquinoline-peroxide combinations, and that these findings cannot be generalized to antimalarials with different mechanisms of action. Note that the aryl amino alcohols, MFQ and LM, were additive with DHA. Since the mechanism of action of MFQ and LM are poorly understood, it is difficult to speculate on a mechanism underlying these interactions.

We agree with the reviewer that while the heme probe may provide some mechanistic insight to explain DHA-quinoline interactions, there is much more to learn about CQ-heme chemistry, particularly within parasites.

The focus of this manuscript was to add a new dimension to considerations about pairings for combination therapies. It is outside the scope of this manuscript to suggest alternative combinations. However, we agree that synergistic combinations would likely be more strategic clinically.

An in vitro setup allows us to eliminate many confounding variables in order to directly assess the impact of partner drugs on DHA activity. However, we agree that in vivo conditions are incredibly more complex, and explicitly state this.

We agree that in the future, modeling studies could provide insight into how antagonism may contribute to real-world efficacy. This is outside the scope of our studies.

**Recommendations for the Authors:**

**Reviewer #1 (Recommendations for the Authors):**
The key weaknesses identified in this manuscript are described in the 'weaknesses' section of the public review. The major one is the inconsistency around the H-FluNox response in the chemical vs biological experiments. I can't think of a simple experiment to resolve this issue, but it is good that this data is openly provided in the manuscript. I believe there could be more discussion to clarify this limitation with the current study, and the conclusions, and particularly the title, should be softened regarding the mechanism of antagonism being based on heme reactivity.

We have softened the title and conclusions to take into account the limitations of our studies.

(1) Please double-check the definitions for isobologram interpretation. In most antimicrobial interaction studies, I see the threshold for antagonism at sumFIC50 of 1.5, or even 2. 1.25 is often interpreted as additive in many studies.

We acknowledge that different studies use various cutoff values. Our interpretations for additive versus antagonistic versus superantagonistic were based not only on mean ΣFIC50 values, but also isobologram shape. For example, the flat isoboles for MFQ-DHA were clearly distinct from the curved isoboles of PPQ-DHA. It is unclear what cutoff value(s) would be most clinically relevant.

(2) For the MFQ-PPQ interaction study, please make it clear that these drugs have very long half-lives (weeks), so the 4 h pulse assay isn't really relevant to their overall activity. It probably shows a slower onset of action, but there is plenty of drug remaining for many days in the clinical scenario, so perhaps the data from the traditional 48h assay is more relevant. The same consideration applies to OZ439, which may impact the interpretation of that data.

We have now included the half-lives of these compounds in the discussion section. Our intent was to use a pulsing format to make these isobolograms comparable with the other assays. It is important to note that pulses can reveal stronger phenotypes that might be missed with traditional methods. Thus, while 48 h assays may better mimic in vivo conditions, they could also mask important phenotypes.

**Reviewer #3 (Recommendations for the Authors):**
I have included most of my concerns in the public review. Below are some additional specific points for consideration:(1) It is expected to include a synergistic combination as a control (e.g., artemisinin + lumefantrine) to contextualize the degree of antagonism observed. The experimental design should show some synergistic profiles in comparison. Adding a few experiments by including a synergistic control is needed.

Both MFQ-DHA and LM-DHA combinations were additive, which provides context for antagonistic combinations. This is now stated in the results section pertaining to Figure 1. We have also included a reference to our previous publication in which we demonstrated that proteasome inhibitor-peroxide combinations are synergistic to additive using this same pulsing format.

(2) Consider in vivo validation or pharmacokinetic/pharmacodynamic modeling to strengthen the translational relevance of the findings when it comes to doses and the IC50 correlations.

We agree that this would be useful to do in future, but it is outside the scope of the current study.

(3) It would be beneficial to include a discussion section on how the findings are generalizable to different *Plasmodium falciparum* genotypes (3D7, Dd2, MRA-1284) and their relevance.

Findings were consistent across three parasite backgrounds depending on PfCRT genotype. This point has been included in the discussion section. The background of these parasites is also provided in Table 1.

(4) Potential evaluation criteria to understand where certain combinations should be reconsidered can be included as a suggestion for the wider audience.

Our in vitro studies suggest that pulsing isobolograms would be a useful assay to include when evaluating combination therapies. While we believe that synergistic combinations would be more strategic than antagonistic combinations, we cannot provide evaluation criteria or make recommendations for reconsidering currently used combinations.

(5) Further elaborate on the mechanistic basis of heme inactivation by quinolines. If data are available, please include more data on the specificity of the process.

Despite our best efforts, we were unable to evaluate quinoline-heme interactions in parasites. Even in vitro, this interaction has remined elusive for decades. We agree that this would be an important future step towards supporting a specific mechanism for quinoline-DHA antagonism.